# GROUP DISTRIBUTIONALLY ROBUST DATASET DISTILLATION WITH RISK MINIMIZATION

**Saeed Vahidian**[1]* **Mingyu Wang**[2]* **Jianyang Gu**[3]* **Vyacheslav Kungurtsev**[4]
**Wei Jiang**[2] **Yiran Chen**[1]

[1] Duke University  [2] Zhejiang University  [3] The Ohio State University
[4] Czech Technical University

## ABSTRACT

Dataset distillation (DD) has emerged as a widely adopted technique for crafting a synthetic dataset that captures the essential information of a training dataset, facilitating the training of accurate neural models. Its applications span various domains, including transfer learning, federated learning, and neural architecture search. The most popular methods for constructing the synthetic data rely on matching the convergence properties of training the model with the synthetic dataset and the training dataset. However, using the empirical loss as the criterion must be thought of as auxiliary in the same sense that the training set is an approximate substitute for the population distribution, and the latter is the data of interest. Yet despite its popularity, an aspect that remains unexplored is the relationship of DD to its generalization, particularly across uncommon subgroups. That is, how can we ensure that a model trained on the synthetic dataset performs well when faced with samples from regions with low population density? Here, the representativeness and coverage of the dataset become salient over the guaranteed training error at inference. Drawing inspiration from distributionally robust optimization, we introduce an algorithm that combines clustering with the minimization of a risk measure on the loss to conduct DD. We provide a theoretical rationale for our approach and demonstrate its effective generalization and robustness across subgroups through numerical experiments.

## 1 INTRODUCTION

Dataset distillation (DD) is a burgeoning area of interest, involving the creation of a synthetic dataset significantly smaller than the real training set yet demonstrating comparable performance on a model (Wang et al., 2018; Sachdeva & McAuley, 2023). This practice has gained prominence in various computation-sensitive applications, providing a valuable means to efficiently construct accurate models (Gu et al., 2023b; Medvedev & D'yakonov, 2021; Xiong et al., 2023). The standard optimization objectives that are used to steer the construction of the synthetic data typically aim to foster either distributional similarity to the training set (Zhao & Bilen, 2023a; Zhao et al., 2023) or similar stochastic gradient descent (SGD) training dynamics as the original dataset (Zhao et al., 2021; Cazenavette et al., 2022). Notably, recent literature suggests that the latter category has proven more successful (Kim et al., 2022; Cazenavette et al., 2023). Intuitively, this success can be attributed to the rationale that, with considerably fewer samples, prioritizing the most relevant information for training and model building becomes more judicious.

In this paper, we aim to address two important practical concerns in DD training. First, it is essential to note that the synthetic dataset might be applied across a wide range of potential circumstances with distinctions from the training phase. Consequently, models trained on the synthetic dataset must exhibit low out-of-distribution error and strong generalization performance. This means the synthetic dataset should be designed to have certain higher-order probabilistic properties, particularly in relation to the model and loss functions. In addition, shifts in circumstances over time (some latent exogenous variables, formally) mean that the population distribution on which the model is trained

---

*Equal contribution.

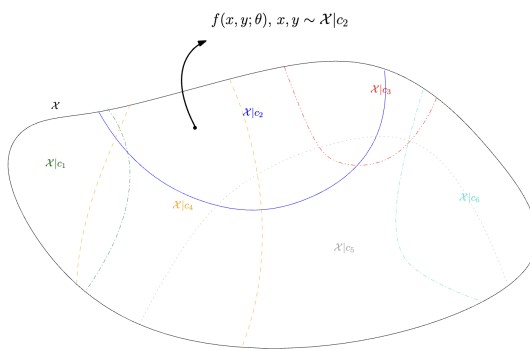

Figure 1: A visual representation of the robust inference task involves the partial partitioning of the population distribution, that is $\mathcal{X}$ across subgroups $\{\mathbf{c}_i\}$. A classifier is considered robust when it demonstrates high performance across the subgroups. As a practical hypothetical example of online learning, at a particular time, a steady stream of samples from $\mathcal{X}|\mathbf{c}_3$ may appear to the classifier. Note that in this case the region of sample space defined by this subgroup is geometrically small, and we can consider that it has a low overall prior density. If this subgroup's behavior is particularly anomalous, a model and any associated distilled dataset trained only on minimizing empirical risk may perform poorly on this subgroup.

may resemble a clustered sample, *i.e.*, a subset of the population. Therefore, it is imperative that the synthetic dataset ensures good coverage across the support of the sample space.

To this end, we propose to enhance the generalization performance and group coverage properties of the distilled dataset using concepts and methods in the field of Distributionally Robust Optimization (DRO) (Lin et al., 2022; Zeng & Lam, 2022; Vilouras et al., 2023). Within DRO, one solves a bilevel optimization problem where the objective is to minimize the loss over a data distribution, subject to the constraint that this distribution is a specific probabilistic distance away from the population distribution:

$$\min_{\theta \in \mathbb{R}^d} \max_{\mathcal{Q} \in \mathbb{Q}}, \ F(\theta, \mathcal{Q}), \ \mathbb{Q} = \{\mathcal{Q} : \mathcal{D}(\mathcal{Q}||\mathcal{P}) \leq \Delta\} \tag{1}$$

where $F$ is the loss function, $\theta$ is the parameter, $\mathbb{Q}$ is the distributional uncertainty set, $\mathcal{D}$ is an f-divergence and $\mathcal{P}$ is the population distribution.

We now present the two desiderata of DD to which we aim to apply the concepts of DRO. First, our goal is to construct a synthetic dataset that is a quality representative of the underlying population distribution. Thus, the consideration of generalization is paramount. Xu & Mannor (2012) present arguments indicating the theoretical equivalence of testing to training error as robustness to perturbations of the data distribution, and the generalization accuracy. Second, a conventional maximum likelihood classifier inherently assigns higher weights to sample ranges with larger prior distributions than to those with lower overall probability density. This inherent bias poses a fundamental risk of underperformance in small population subgroups. Furthermore, in online inference, it is common to observe alterations in the overall prior distribution, particularly in terms of subgroups. DRO has been conjectured and observed to mitigate this issue, both generically (Duchi & Namkoong, 2021) and with explicit quantification of subgroups in the distributional uncertainty set (Oren et al., 2019), where subgroups of topics are considered for training language models.

Moreover, this permits flexibility as far as easily modularized methods to specific concerns. Indeed, domain expertise in regards to the properties of the population distribution has been shown to assist in generalization– (Hu et al., 2018) indicates that a more precisely defined distributional uncertainty set, focusing on specific collections of probability distributions from subsamples of the population rather than encompassing all possibilities in a ball, yields improved generalization performance.

In a formal sense, we define that each subsample involved in an inference task is drawn from a union of closed, convex, connected subsets of the population distribution. An intuitive representation of the envisioned scenario is illustrated in Figure 1.

Inspired by these concepts, this paper introduces a Double-DRO-based DD procedure designed specifically to address these two concerns. Fundamentally we employ DRO at two levels to promote different advantages that we shall witness in the experimental results. 1) We leverage the loss across clusters of the latent variables to ensure group robustness. 2) We approximately solve a DRO problem internally in order to promote better within-group generalization, using a risk measure as a proxy for a DRO solution.

Our paper continues below as follows. Next, we present our algorithm, we analyze the optimization problem defining the notion of generalization of interest, and argue how our method effectively targets this criterion in Section 2. We give the mathematical analysis in Section 3. In Section 4 we present numerical results validating the superior overall and subgroup generalization performance on standard machine learning benchmarks. Finally, we conclude this work in Section 5. Related works are reviewed in the Appendix in Section E.

## 2 ALGORITHM

In this Section we describe our main procedure, in which we incorporate two techniques informed by DRO in order to improve both generic generalization performance and group distributional coverage.

Let $\mathcal{S} = \{x_S, y_S\}$ be the synthetic distilled dataset, where $([x_S]_i, [y_S]_i)$ with $i = 1, ..., |\mathcal{S}|$ is an individual input and label pair. Let $\mathcal{T} = \{x_T, y_T\}$ be the real training set where $([x_T]_i, [y_T]_j)$, $j = 1, ..., |\mathcal{T}|$ is an individual input and label pair with $|\mathcal{T}| \gg |\mathcal{S}|$. The target of dataset distillation is to optimize the synthetic dataset $\mathcal{S}$ so that a network with parameter $\theta$ can achieve similar performance compared with that attained on $\mathcal{T}$. For simplicity, we define the input and label pair as $z = (x, y)$. And $F(\theta; z)$ is the full loss function to optimize parameter $\theta$ on data $z$, which is typically cross entropy loss in classification tasks.

Matching-based dataset distillation methods usually involve imitating certain training characteristics of the real training set, such as training gradients, feature distribution, training trajectory, *etc* (Zhao et al., 2021; Zhao & Bilen, 2023a; Cazenavette et al., 2022). Take gradient matching as an example. At each iteration, the training gradient is extracted for the synthetic data and the real data, based on the same network. The gradient difference is set as the objective of optimizing synthetic data. Considering that diverse supervision gradients from different training stages should be provided to enhance the consistency throughout a model training process, the network is simultaneously updated with the synthetic data optimization. Previous methods utilize the same loss function $F(\theta; z)$ to update the network. However, in this work, we propose to incorporate DRO at this stage for more robust network optimization as well as distillation supervision.

Specifically, we wish to solve a DRO locally in order to promote solutions that are particularly suitable for generalization. However, as the dimensionality and desired training size in our settings of interest present intractability for DRO, instead of solving the full DRO we use a proxy in the form of a Conditional Value at Risk (CVaR), which we define below. In the theoretical analysis, we describe the known correspondence between CVaR and DRO.

In our algorithm, each iteration begins with subsampling the training dataset $\bar{\mathcal{T}} \subset \mathcal{T}$ and then clustering them based on their nearest synthetic data point. That is for all $i \in [|\mathcal{S}|]$ the set $\mathbf{c}_i$ contains the elements of $\bar{\mathcal{T}}$ that are closer to $z_i$ than to any other point in $\mathcal{S}$. The following criterion optimization problem is solved for $\theta$, obtaining the set of parameters that solves for minimax loss across the subgroups defined by the clusters $\{\mathbf{c}_i\}$

$$\min_\theta \left[ \frac{1}{|\mathcal{S}|} \sum_i \texttt{CVaR}[F(\theta; \mathbf{c}_i)] + \max_i \texttt{CVaR}[F(\theta; \mathbf{c}_i)] \right]. \tag{2}$$

By considering the clusters $\mathbf{c}_i$ separately, especially including the $\max_i$ term in the objective, we robustify the performance with respect to the different clusters. The synthetic dataset is meant to represent the training set, and with clustering, we ensure the entire support of the training distribution is represented by some sample $z_i \in \mathcal{S}$. This yields our effort to ensure the group distributional robustness. We intend that both the worst case and the average performance across the sub-groups are minimized.

In addition, to promote generalization broadly speaking, note that rather than the expectation, we use a risk measure, as far as statistically agglomerating the loss over the training data. We set $\alpha$ to be some tail probability. Risk measures enable one to minimize one-sided tail behavior. The operator denoting the Conditional Value at Risk, $\texttt{CVaR}$, is defined, with respect to the empirical distribution of data points within $\mathbf{c}_i$:

$$\texttt{CVaR}[F(\theta; \mathbf{c}_i)] := -\frac{1}{\alpha} \left\{ \frac{1}{|\mathbf{c}_i|} \sum_{z_t \in \mathbf{c}_i} F(\theta; z_t) \mathbf{1}[F(\theta; z_t) \le f_\alpha] \right.$$
$$\left. + f_\alpha (\alpha - \frac{1}{|\mathbf{c}_i|} \sum_{z_t \in \mathbf{c}_i} \mathbf{1}[F(\theta; z_t) \le f_\alpha]) \right\} \tag{3}$$

---

**Algorithm 1** Robust Dataset Distillation

---

**Input:** Real training set $\mathcal{T}$, synthetic set $\mathcal{S}$, network with parameter $\theta$, distilling objective $L(\mathcal{S})$.

**Execute:**
    **While** not converged:
        Subsample the training set $\bar{\mathcal{T}} \subset \mathcal{T}$.
        Cluster $\bar{\mathcal{T}}$ by the distillation set, i.e. define, for all $t \in [|\bar{\mathcal{T}}|]$:
            $\mathcal{C}(z_t) = \arg\min_i \|z_t - [\mathcal{S}]_i\|^2$ and
            $\mathbf{c}_i = \{z_t \in \bar{\mathcal{T}} : \mathcal{C}(z_t) = i\}$.
        Solve the optimization problem in equation 2 to obtain $\theta^*$.
        Optimize synthetic set $\mathcal{S}$ with $L(\mathcal{S})$ based on the optimized parameter $\theta^*$.

---

with the quantity $f_\alpha$ defined as, with $|\mathbf{c}_i|$ denoting the size of the training set in cluster $\mathbf{c}_i$,

$$f_\alpha := \min \left\{ f \in \mathbb{R} : \frac{1}{|\mathbf{c}_i|} \sum_{z_t \in \mathbf{c}_i} \mathbf{1}[F(\theta; z_t) \leq f_\alpha] \geq \alpha \right\} \tag{4}$$

The choice of a CVaR weighing of the loss informed by DRO-associated theoretical work aims to improve test error accuracy rather than merely minimizing training error. Additionally, by clustering and weighting the balanced loss, Group DRO is applied across the connected components of the sample space distribution, which defines the population.

After updating the network with Group DRO, the synthetic dataset $\mathcal{S}$ will be optimized based on matching metrics, denoted by $L(\mathcal{S})$, where existing matching-based DD methods can be broadly plugged in. This makes the algorithm modular to any choice of DD procedure. The algorithm is shown in Alg. 1. As $L(\mathcal{S})$ involves a risk function of the loss $F(\theta; z)$, averaged across a set of partitions of $\mathcal{S}$, the supervision helps enhance the distributional robustness of the distilling process. Thereby, the distilled dataset can obtain superior generalization performance across different domain shifts, as well as better group coverage for more practical usefulness. We proceed to iterate between an update to the parameter based on solving equation 2 and an update to the synthetic dataset $\mathcal{S}$ with our tailored DD procedure until the procedure reaches a fixed point, wherein the two do not significantly change.

## 3 ANALYSIS

In this Section we describe the formal optimization problem being solved as well as its convergence and statistical properties. The focus will be on justifying why we expect the procedures defined in the Algorithm to improve (group) generalization, through foundational results in DRO theory.

Let $\mathcal{Q}$ be a *partition* of the population distribution $\mathcal{D}$. As such, we can describe the problem of group coverage at the point of inference as yielding comparable performance across elements of $\mathcal{Q}$. To this end, we denote the support of the population distribution as $D = \text{supp}(\mathcal{D})$ and we assume that it is compact. Let us define,

$$\mathbb{Q} = \{\mathcal{Q}_i\}, \text{ with } Q_i = \text{supp}(\mathcal{Q}_i), \bigcup_{i=1}^{q} Q_i = D, \underline{s} \leq \lambda(Q_i) \leq \bar{s} \tag{5}$$

for some $\bar{s} > \underline{s} > 0$ and every $Q_i$ is itself a union of compact, convex, and connected sets. Here $\lambda$ is integration with respect to the Lebesgue measure. Recall that we denote the synthetic dataset by $\mathcal{S}$ with $|\mathcal{S}| = S$.

Note the assumption that $\mathcal{D}$ is finite. In addition, we assume a certain regularity of the loss function. Formally, we state the following:

**Assumption 3.1.** The population distribution $\mathcal{D}$ has bounded support, i.e., $\lambda(D) < \infty$. The loss $F$ is Lipschitz continuously differentiable with respect to the first argument (the parameters) and continuous with respect to the second argument (the input features and labels). Finally, the partitions are not probabilistically small, i.e., there exists $p_{q_0}$ such that $\mathbb{P}_{\mathcal{D}}[A \in \mathcal{Q}_i] \geq p_{q_0}$ for all $i$.

Additionally, let's explore the asymptotic learning regime when examining the problem, where the entire population set could be sampled in the limit.

**Assumption 3.2.** For all $\mathcal{Q} \in \mathbb{Q}$, consider the asymptotic online learning limit,

$$\lim_{t \to \infty} \sup \left( \text{supp}(\mathcal{Q}) \cap \bar{\mathcal{T}}_t \right) = \text{supp}(\mathcal{Q}) \tag{6}$$

where $\bar{\mathcal{T}}_t$ is the training set sampled at iteration $t$.

Informally, group coverage corresponds to enforcing adequate performance for prediction using the trained model regardless of what portion of the population set is taken. Essentially, at a particular future instant, the learner could be expected to classify or predict a quantity for some small subpopulation cohort that may appear as a subsample at the time of online inference.

Let us formally articulate the optimization problem of interest, as outlined in the Introduction, as follows

$$\min_{\mathcal{S}} \max_{\mathcal{Q} \in \mathbb{Q}} \mathbb{E}[F(\theta^*, \mathcal{Q})] \quad \text{s.t. } \theta^* \in \arg\min_\theta F(\theta, \mathcal{S}). \tag{7}$$

Let us briefly consider the standard circumstance by which Algorithm 1 converges. By considering each iteration as two players' best response to a cooperative Nash game, we can ensure convergence asymptotically of iteratively solving the two optimization problems, as given in Nash (1953).

**Theorem 3.3.** *Under the circumstance by which the Morse-Saard condition holds (Souček & Souček, 1972) and so the optimal set $\{\mathcal{S}^*(\theta)\}, \{\theta^*(\mathcal{S})\}$ is compact (possibly finite) for all $\theta, \mathcal{S}$, then Algorithm 2 converges to a fixed point of equation 7.*

A remark on the time complexity of the Algorithm. Using CVaR instead of a sample average only takes a constant multiple of operations on the subsample. Clustering itself can be done polynomial in the number of samples. Since the properties of the model and loss function, that is, nonconvex and nonsmooth, are fundamentally unchanged, there is no change to the iteration complexity.

Now we will discuss generalization and DRO. Recall that there are two layers of generalization: on the one hand, we are solving a problem robust with respect to the choice of $\mathcal{Q} \in \mathbb{Q}$, and on the other hand, we are considering the population error rather than an empirical loss.

We posit that the computation of a gradient estimate employs a conventional Stochastic Approximation procedure to address *some* bilevel optimization problem for $\mathcal{S}$. Thus we do not address the convergence guarantees (that is, asymptotic stationarity and convergence rate) of the training procedure itself but study the properties of the associated optimization problems and their solutions.

Accordingly, our emphasis is on scrutinizing the properties associated with the criterion,

$$\min_\theta \max_{\mathcal{Q} \in \mathbb{Q}} \mathbb{E}_Q[F(\theta, \mathcal{Q})]. \tag{8}$$

that is ultimately used to steer the synthetic dataset, as the DD task is finding a dataset on whose associated $\theta$-minimizer also minimizes this quantity. In the analysis, we shall argue about the validity of these criteria as far ensuring the generalization as well as group robustness properties of the solution of equation 7.

To begin with, we rewrite equation 7 as:

$$\min_{\mathcal{S}} \max_{\mathcal{Q} \in \mathbb{Q}} \max_{\mathcal{Q}' \in \bar{\mathcal{Q}}} \mathbb{E}[F(\theta^*, \mathcal{Q}')] \quad \text{s.t.} \quad \begin{aligned} \theta^* &\in \arg\min_\theta F(\theta, \mathcal{S}) \\ \bar{\mathcal{Q}} &= \{\mathcal{Q}' : I(\mathcal{Q}', \mathcal{S} \cup \mathcal{Q}_N) \le r\} \end{aligned} \tag{9}$$

where $r > 0$ is some bound and we replace the inner problem to be evaluated on the entire training dataset with a data-driven DRO. In the appendix we present several results in the literature that indicate how the DRO to an empirical risk minimization problem exhibits generalization guarantees and hence is a valid auxiliary criterion for minimizing equation 7.

## 3.1 Large Deviations and Solving the bilevel DRO

To justify the clustering and risk measure minimization algorithm as an appropriate procedure for solving equation 9, we apply some theoretical analysis on the relationship between DRO and

Large Deviations Principles (LDP). LDPs (*e.g.*, Deuschel & Stroock (2001)) define an exponential asymptotic decay of the measure of the tails of an empirical distribution with respect to a sample size.

The particular, Large Deviations Principle (LDP) capturing out-of-sample disappointment, which is of interest to us, is defined as follows:

$$\lim_{N\to\infty} \sup \frac{1}{N} \log \mathbb{P}_{\mathcal{Q}}^{\infty} \left( F(\theta^*(\mathcal{S}), \mathbb{P}_{\mathcal{Q}}) > F(\theta^*(\mathcal{S}), \mathcal{S} \cup \mathcal{Q}_N) \right) \leq -r, \; \forall \mathcal{Q} \in \mathbb{Q} \tag{10}$$

for some $r > 0$. This states that for all partitions $\mathcal{Q}$ of the population space partition $\mathbb{Q}$, there is an exponential asymptotic decay of the probability of one-sided sample error (i.e., samples of size $N$ from the population $\mathbb{P}_{\mathcal{Q}}$) relative to the computed loss on the synthetic and training data points $\mathcal{Q}_N$, as the number of training data points grows asymptotically.

Now, we present the following Proposition, whose proof is in the appendix,

**Proposition 3.4.** *Algorithm equation 2 satisfies, under Assumptions 3.1 and 3.2, the LDP equation 10.*

Next, we relate Algorithm equation 2 together with 3.1 to the DRO problem equation 9.

## 3.2 LARGE DEVIATIONS AND DISTRIBUTIONALLY ROBUST OPTIMIZATION

Applying Assumption 3.2, the LDP, and the continuity of $F$ with respect to the data, Assumption 3.1 (see also the proof of Lemma 1 as well as Example 3 in Duchi & Namkoong (2021)) we can bound the quantity:

$$\mathbb{P}(F(\theta^*, q) > F(\theta^*, \mathcal{S} \cup \mathcal{Q}_N) \tag{11}$$

for $q \in \mathcal{Q}'$, with $\mathcal{D}(\mathcal{Q}_N, \mathcal{Q}') \leq r$ with a sufficient step-size.

This implies that one can bound the objective value of the DRO problem of interest equation 9:

$$\min_{\mathcal{S}} \max_{\mathcal{Q} \in \mathbb{Q}} \max_{\mathcal{Q}' \in \bar{\mathcal{Q}}} \mathbb{E}[F(\theta^*, \mathcal{Q}')] \quad \text{s.t.} \quad \begin{aligned} &\theta^* \in \arg\min_\theta F(\theta, \mathcal{S}) \\ &\bar{\mathcal{Q}} = \{ \mathcal{Q}' : I(\mathcal{Q}', \mathcal{S} \cup \mathcal{Q}_N) \leq r \} \end{aligned} \tag{12}$$

i.e., there is some small $C > 0$ such that,

$$|O(\theta^*, S^*) - \hat{O}(\hat{\theta}, \hat{S})| \leq C \tag{13}$$

where $O$ and $\hat{O}$ refer to the optimal value of the data driven DRO equation 9 and the approximation given by Algorithm 1 and 2.

## 3.3 DISTRIBUTIONALLY ROBUST OPTIMIZATION AND SUBGROUP COVERAGE

Let us return to equation 7:

$$\min_{\mathcal{S}} \max_{\mathcal{Q} \in \mathbb{Q}} \mathbb{E}[F(\theta^*, \mathcal{Q})] \quad \text{s.t. } \theta^* \in \arg\min_\theta F(\theta, \mathcal{S})$$

We have established that our Algorithm approximately solves a DRO which approximately bounds the population loss. Now consider the outer DRO itself. The same theory regarding DRO and LDPs can now be applied, but now to yield a stronger result, since we are evaluating the full (test) loss. Indeed the data being sampled are simply $\mathcal{Q}_\omega \in \mathbb{Q}$, that is, some i.i.d. selection $\omega$ over the finite set of partitions. Since the entire subpopulation is taken, each estimator is constructed with the full population error, and w.p. 1 the entire set $\mathbb{Q}$ is sampled for finite $N$.

We can directly apply Theorem 7 in the work of Van Parys et al. (2021) to deduce that the result satisfies:

$$\lim_{N\to\infty} \sup \frac{1}{N} \log \mathbb{P}_\omega^\infty \left( F(\theta^*, Q_\omega) > F(\theta^*, \mathcal{S}) \right) \leq -r. \tag{14}$$

Thus by the Kolmogorov 0-1 principle we have that,

$$F(\theta^*, \mathcal{Q}) \leq F(\theta^*, \mathcal{S}),$$

for all $\mathcal{Q}$. This guarantees the quality of $\mathcal{S}$ in ensuring group robust guarantees on the loss.

Table 1: Top-1 test accuracy on robustness testing sets. "Gain" denotes the performance gain of applying our proposed method. Except for absolute values, the relative fluctuation compared with the standard case is also reported as subscripts. The experiments are conducted under the IPC setting of 10. $\mathcal{R}$ indicates the proposed robust dataset distillation method applied. The better results between baseline and the proposed method are marked in **bold**.

| Dataset | Setting | Random | IDC | IDC$^{\mathcal{R}}$ | Gain | GLaD | GLaD$^{\mathcal{R}}$ | Gain |
|---|---|---|---|---|---|---|---|---|
| CIFAR-10 | Standard | $37.2_{\pm 0.8}$ | $67.5_{\pm 0.5}$ | $\mathbf{68.6}_{\pm 0.2}$ | 1.1 | $46.7_{\pm 0.6}$ | $\mathbf{50.2}_{\pm 0.5}$ | 3.5 |
| | Cluster-min | $31.4_{\pm 0.9}$ | $63.3_{\pm 0.7}\downarrow 4.2$ | $\mathbf{65.0}_{\pm 0.6}\downarrow 3.6$ | $1.7_{\uparrow 0.6}$ | $40.2_{\pm 1.2}\downarrow 6.5$ | $\mathbf{46.7}_{\pm 0.9}\downarrow 3.5$ | $6.5_{\uparrow 3.0}$ |
| | Noise | $35.4_{\pm 1.2}$ | $57.2_{\pm 1.1}\downarrow 10.3$ | $\mathbf{59.4}_{\pm 1.0}\downarrow 9.2$ | $2.2_{\uparrow 1.1}$ | $44.1_{\pm 0.6}\downarrow 2.5$ | $\mathbf{49.5}_{\pm 0.5}\downarrow 0.7$ | $5.4_{\uparrow 1.8}$ |
| | Blur | $29.4_{\pm 0.6}$ | $48.3_{\pm 0.9}\downarrow 19.2$ | $\mathbf{50.5}_{\pm 0.7}\downarrow 18.1$ | $2.2_{\uparrow 1.1}$ | $36.9_{\pm 0.6}\downarrow 9.8$ | $\mathbf{39.0}_{\pm 0.6}\downarrow 11.2$ | $2.1_{\downarrow 1.4}$ |
| | Invert | $9.5_{\pm 1.0}$ | $25.6_{\pm 0.5}\downarrow 41.9$ | $\mathbf{26.5}_{\pm 0.6}\downarrow 42.1$ | $0.9_{\downarrow 0.2}$ | $10.6_{\pm 1.1}\downarrow 36.1$ | $\mathbf{13.0}_{\pm 1.2}\downarrow 37.2$ | $2.4_{\downarrow 1.1}$ |
| ImageNet-10 | Standard | $46.9_{\pm 0.7}$ | $72.8_{\pm 0.6}$ | $\mathbf{74.6}_{\pm 0.9}$ | 1.8 | $50.9_{\pm 0.9}$ | $\mathbf{55.2}_{\pm 1.1}$ | 4.3 |
| | Cluster-min | $31.2_{\pm 1.0}$ | $61.4_{\pm 1.0}\downarrow 11.4$ | $\mathbf{65.7}_{\pm 0.5}\downarrow 8.9$ | $4.3_{\uparrow 2.5}$ | $34.9_{\pm 0.9}\downarrow 16.0$ | $\mathbf{47.1}_{\pm 1.2}\downarrow 8.1$ | $12.2_{\uparrow 7.9}$ |
| | Noise | $42.6_{\pm 0.8}$ | $65.8_{\pm 0.8}\downarrow 7.0$ | $\mathbf{68.8}_{\pm 0.9}\downarrow 5.8$ | $3.0_{\uparrow 1.2}$ | $48.6_{\pm 0.7}\downarrow 2.3$ | $\mathbf{53.9}_{\pm 0.8}\downarrow 1.3$ | $5.3_{\uparrow 1.0}$ |
| | Blur | $45.5_{\pm 1.1}$ | $71.9_{\pm 0.9}\downarrow 0.9$ | $\mathbf{74.1}_{\pm 1.1}\downarrow 0.5$ | $2.2_{\uparrow 0.4}$ | $47.9_{\pm 0.9}\downarrow 3.0$ | $\mathbf{54.6}_{\pm 0.9}\downarrow 1.6$ | $6.7_{\uparrow 1.4}$ |
| | Invert | $21.0_{\pm 0.6}$ | $27.8_{\pm 0.7}\downarrow 45.0$ | $\mathbf{30.3}_{\pm 0.8}\downarrow 44.3$ | $2.5_{\uparrow 0.7}$ | $17.0_{\pm 1.0}\downarrow 33.9$ | $\mathbf{21.6}_{\pm 0.7}\downarrow 33.6$ | $4.6_{\uparrow 0.3}$ |

Table 2: Top-1 test accuracy on standard testing sets. $^{\dagger}$ indicates the result is reported based on our runs. $\mathcal{R}$ indicates the proposed robust dataset distillation method applied on the baseline. The better results between baseline and the proposed method are marked in **bold**.

| Dataset | IPC | Random | DSA | DM | KIP | IDC | IDC$^{\mathcal{R}}$ | GLaD | GLaD$^{\mathcal{R}}$ |
|---|---|---|---|---|---|---|---|---|---|
| SVHN | 1 | 14.6 | 27.5 | 24.2 | 57.3 | $68.5_{\pm 0.9}$ | $\mathbf{68.9}_{\pm 0.4}$ | $32.5_{\pm 0.5}{}^{\dagger}$ | $\mathbf{35.7}_{\pm 0.3}$ |
| | 10 | 35.1 | 79.2 | 72.0 | 75.0 | $87.5_{\pm 0.3}$ | $\mathbf{88.1}_{\pm 0.3}$ | $68.2_{\pm 0.4}{}^{\dagger}$ | $\mathbf{72.5}_{\pm 0.4}$ |
| | 50 | 70.9 | 84.4 | 84.3 | 80.5 | $90.1_{\pm 0.1}$ | $\mathbf{90.8}_{\pm 0.4}$ | $71.8_{\pm 0.6}{}^{\dagger}$ | $\mathbf{76.6}_{\pm 0.3}$ |
| CIFAR-10 | 1 | 14.4 | 28.7 | 26.0 | 49.9 | $50.6_{\pm 0.4}{}^{\dagger}$ | $\mathbf{51.3}_{\pm 0.3}$ | $28.0_{\pm 0.8}{}^{\dagger}$ | $\mathbf{29.2}_{\pm 0.8}$ |
| | 10 | 37.2 | 52.1 | 53.8 | 62.7 | $67.5_{\pm 0.5}$ | $\mathbf{68.6}_{\pm 0.2}$ | $46.7_{\pm 0.6}{}^{\dagger}$ | $\mathbf{50.2}_{\pm 0.5}$ |
| | 50 | 56.5 | 60.6 | 65.6 | 68.6 | $74.5_{\pm 0.1}$ | $\mathbf{75.3}_{\pm 0.5}$ | $59.9_{\pm 0.9}{}^{\dagger}$ | $\mathbf{62.5}_{\pm 0.7}$ |
| ImageNet-10 | 1 | $23.2^{\dagger}$ | $30.6^{\dagger}$ | $30.2^{\dagger}$ | - | $54.4_{\pm 1.1}{}^{\dagger}$ | $\mathbf{58.2}_{\pm 1.2}$ | $33.5_{\pm 0.9}{}^{\dagger}$ | $\mathbf{36.4}_{\pm 0.8}$ |
| | 10 | 46.9 | 52.7 | 52.3 | - | $72.8_{\pm 0.6}$ | $\mathbf{74.6}_{\pm 0.9}$ | $50.9_{\pm 1.0}{}^{\dagger}$ | $\mathbf{55.2}_{\pm 1.1}$ |

## 4 NUMERICAL RESULTS

In this section we present numerical results to validate the efficacy of the proposed robust dataset distillation method. Implementation details are listed in Sec. H.

**Results on Robustness Settings** We first show the notable advantage offered by our proposed method that is the robustness against various domain shifts. This property is assessed through multiple protocols. Firstly, as suggested before, we present validation results on different partitions of the testing set. A clustering process is conducted to divide the original testing set into multiple sub-sets. We test the performance on each of them and report the worst accuracy among the sub-sets to demonstrate the robustness of distilled data, denoted as "Cluster-min" in Tab. 1. In the sub-scripts, the performance drop compared with the standard case is reported. Several key observations emerge from the experiment results. (1) Compared with random images, the Cluster-min accuracy of baseline DD methods exhibits improvement alongside the standard performance. It suggests that by condensing the knowledge from original data into informative distilled samples, DD methods contribute to enhanced data robustness. (2) Compared with CIFAR-10, the performance gap between the standard case and the worst sub-cluster on ImageNet-10 is more pronounced. This discrepancy can be attributed to a higher incidence of ID-unrelated interruptions in ImageNet-10, resulting in larger domain-shifts between sub-clusters and the original distribution. This finding aligns with the observation in Tab. 2. (3) With our proposed robust method applied, not only is the cluster-min performance improved, but the performance drop from the standard case is also significantly mitigated compared with baselines. It suggests exceptional overall generalization and robustness conferred by our method.

Furthermore, we provide testing results in Tab. 1 on truncated testing sets, simulating scenarios where the testing set exhibits more substantial domain shifts compared with the training data. We employ three data truncation means, including the addition of Gaussian noise, application of blur effects, and

Table 3: Top-1 test accuracy on subpopulation shift benchmarks MetaShift and ImageNetBG. All the results are reported based on the average of 5 runs. $\mathcal{R}$ indicates the proposed robust dataset distillation method applied. The better results between baseline and the proposed method are marked in **bold**.

| Metric | MetaShift | | ImageNetBG | |
| --- | --- | --- | --- | --- |
| | GLaD | GLaD$^{\mathcal{R}}$ | GLaD | GLaD$^{\mathcal{R}}$ |
| Average Accuracy | $58.6_{\pm 2.3}$ | $\mathbf{62.2}_{\pm 1.2}$ | $41.7_{\pm 1.5}$ | $\mathbf{45.5}_{\pm 1.1}$ |
| Worst-group Accuracy | $51.3_{\pm 1.8}(\downarrow 7.3)$ | $\mathbf{57.0}_{\pm 1.0}(\downarrow 5.2)$ | $32.2_{\pm 1.6}(\downarrow 9.5)$ | $\mathbf{38.6}_{\pm 1.0}(\downarrow 6.9)$ |

Table 4: (a) Cross-architecture generalization performance comparison. The experiment is conducted on ImageNet-10 under 10 IPC settings. The distilling architecture for IDC is ResNet-10, while for GLaD is ConvNet-5. $\mathcal{R}$ indicates the proposed robust dataset distillation method applied. (b) Ablation study on CVaR loss. The experiment is conducted on CIFAR-10 and ImageNet-A under 10 IPC. The baseline model is GLaD. "CE" denotes cross entropy loss, and "CL-min" refers to the worst sub-cluster accuracy. "aCVaR" and "mCVaR" refer to average and maximum CVaR loss, respectively. The best results are marked in **bold**.

(a)

| Method | Architecture | | | | |
| --- | --- | --- | --- | --- | --- |
| | Conv | Res10 | Res18 | ViT | VGG11 |
| IDC | $71.9_{\pm 0.8}$ | $72.8_{\pm 0.5}$ | $70.8_{\pm 1.0}$ | $55.2_{\pm 1.2}$ | $64.5_{\pm 0.6}$ |
| IDC$^{\mathcal{R}}$ | $\mathbf{72.6}_{\pm 0.6}$ | $\mathbf{74.6}_{\pm 0.7}$ | $\mathbf{72.7}_{\pm 0.8}$ | $\mathbf{56.4}_{\pm 1.1}$ | $\mathbf{65.6}_{\pm 0.5}$ |
| GLaD | $48.2_{\pm 0.7}$ | $50.9_{\pm 0.5}$ | $51.2_{\pm 0.4}$ | $36.8_{\pm 0.9}$ | $44.2_{\pm 1.0}$ |
| GLaD$^{\mathcal{R}}$ | $\mathbf{51.9}_{\pm 0.7}$ | $\mathbf{55.2}_{\pm 0.6}$ | $\mathbf{53.6}_{\pm 0.5}$ | $\mathbf{39.2}_{\pm 0.9}$ | $\mathbf{46.3}_{\pm 0.8}$ |

(b)

| Loss | | | CIFAR-10 | | ImageNet-A | |
| --- | --- | --- | --- | --- | --- | --- |
| CE | aCVaR | mCVaR | Acc | CL-min | Acc | CL-min |
| ✓ | - | - | $46.7_{\pm 0.6}$ | $40.2_{\pm 0.8}$ | $53.9_{\pm 0.6}$ | $40.5_{\pm 0.7}$ |
| ✓ | ✓ | - | $49.1_{\pm 0.7}$ | $45.3_{\pm 1.0}$ | $56.4_{\pm 0.5}$ | $43.9_{\pm 0.8}$ |
| ✓ | - | ✓ | $47.9_{\pm 0.8}$ | $44.2_{\pm 0.9}$ | $56.1_{\pm 0.8}$ | $43.7_{\pm 0.9}$ |
| ✓ | ✓ | ✓ | $\mathbf{50.2}_{\pm 0.6}$ | $\mathbf{46.7}_{\pm 0.8}$ | $\mathbf{57.5}_{\pm 0.7}$ | $\mathbf{45.8}_{\pm 0.5}$ |

inversion of image colors. The conclusions drawn from truncated testing sets align with those from partitioned testing sets. While dataset distillation generally contributes to improved data robustness, the introduction of CVaR loss further amplifies the trend. In this analysis with a higher resolution to capture more details, we observe that truncation on ImageNet has a smaller impact compared with CIFAR-10. Generally, the improvement by our proposed method on truncated performance is also larger than that on standard testing sets, especially on ImageNet-10. The increased advantage on robustness settings further validates that the proposed method not only elevates overall accuracy but significantly fortifies the robustness of distilled data.

**Subpopulation Shift Experiments**   In addition to the previous standard classification tasks, we also extend the method to subpopulation shift benchmarks MetaShift and ImageNetBG (Yang et al., 2023; Liang & Zou, 2022; Xiao et al., 2021). The two benchmarks consider the spurious correlation and attribute generalization problems, respectively. We use GlaD as the baseline method to distill 50 images for each class. Subsequently, we conduct the standard evaluation as in the other datasets, with the results reported in Table 3. The proposed robust dataset distillation algorithm not only improves the average accuracy, but also yields a smaller worst-group accuracy margin compared with the baseline method. The results further suggest that RDD can enhance the robustness of distilled data for datasets with subpopulation shift issues.

**Results on Standard Benchmark**   We then evaluate our proposed method on standard testing sets, including SVHN (Sermanet et al., 2012), CIFAR-10 (Krizhevsky et al., 2009), ImageNet-10 (Deng et al., 2009), and Tiny-ImageNet (Deng et al., 2009) in Table 2. The ImageNet-10 split follows the configuration outlined in IDC (Kim et al., 2022). We use IDC (Kim et al., 2022) and GLaD (Cazenavette et al., 2023) as baselines in this section, representing distilling at the pixel level and the latent level, respectively. Additionally, the validation results are also compared with DSA, DM, and KIP (Zhao & Bilen, 2021; 2023b; Nguyen et al., 2021).

The incorporation of CVaR loss consistently enhances performance across all scenarios. Notably, under the IPC of 10, facilitated by the supervision from segregated sub-clusters, our proposed method demonstrates the most substantial performance improvement over baselines. In the case of 1 IPC, where only a single synthetic sample is available for sub-cluster construction, the optimization comes back to a DRO problem. And minimizing CVaR still captures tail risk and helps improve the validation performance, despite the absence of guidance from multiple sub-clusters. Our proposed

Table 5: Top-1 robustness evaluation on IDM (CIFAR-10) and GLaD (TinyImageNet).

| Setting | Method | | | |
|---|---|---|---|---|
| | IDM | IDM$^{\mathcal{R}}$ | GLaD | GLaD$^{\mathcal{R}}$ |
| Acc | $67.5_{\pm 0.2}$ | $\mathbf{68.1}_{\pm 0.1}$ | $24.9_{\pm 0.5}$ | $\mathbf{26.8}_{\pm 0.6}$ |
| Cluster-min | $62.1_{\pm 0.3}$ | $\mathbf{63.2}_{\pm 0.2}$ | $20.5_{\pm 0.8}$ | $\mathbf{22.6}_{\pm 0.6}$ |
| Noise | $61.8_{\pm 0.5}$ | $\mathbf{62.8}_{\pm 0.6}$ | $22.1_{\pm 0.6}$ | $\mathbf{24.4}_{\pm 0.7}$ |
| Blur | $54.5_{\pm 0.5}$ | $\mathbf{56.2}_{\pm 0.4}$ | $20.8_{\pm 0.3}$ | $\mathbf{23.2}_{\pm 0.5}$ |
| Invert | $20.3_{\pm 0.5}$ | $\mathbf{21.5}_{\pm 0.7}$ | $10.2_{\pm 1.2}$ | $\mathbf{11.7}_{\pm 0.9}$ |

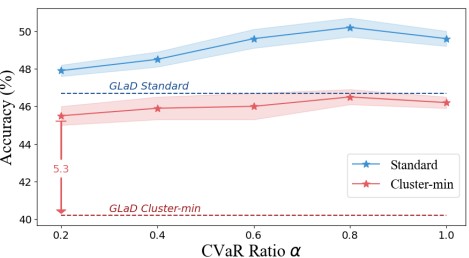

Figure 2: Analysis on CVaR ratio $\alpha$.

method is especially effective on ImageNet-10, characterized by finer class divisions and more substantial intra-class variation compared with CIFAR-10. On ImageNet-10 we achieve an average top-1 accuracy gain of 3.1%, further elevating the state-of-the-art baseline in DD methods. More results are presented in the Appendix Section D.

**Cross-architecture Generalization** In addition to enhanced robustness against domain-shifted data, the incorporation of CVaR loss also yields better cross-architecture generalization capabilities. We assess multiple network architectures including ConvNet-5, ResNet, ViT (Dosovitskiy et al., 2020) and VGG11 (Simonyan & Zisserman, 2014), and compare the performance with and without our proposed robust method in Table 4a. Despite the different distilling architectures employed in IDC and GLaD, both methods achieve their highest accuracy on ResNet-10. Notably, the proposed robust distillation method consistently enhances performance across all architectures, showcasing remarkable cross-architecture generalization capabilities.

**Ablation Study** We conduct an ablation study on the incorporation of CVaR loss in Table 4b. In addition to CIFAR-10, we also report results on the ImageNet-A sub-set according to the setting in the work of Cazenavette et al. (2023). Both accuracy on the standard testing set and the worst sub-set accuracy "Cluster-min" are presented. Our focus is primarily on two aspects of utilizing the CVaR loss, *i.e.* the maximum value and average value of CVaR losses across all sub-clusters. The CVaR loss operates as a complement to the standard cross-entropy optimization. Hence the baseline case, involving only cross entropy loss, mirrors the performance of GLaD. Compared with the maximum value, the average CVaR loss proves more effective in enhancing the validation performance when applied independently. While both average and maximum CVaR loss yield considerable improvement over the baseline, their combined application further fortifies the robustness of the distilled data, which is selected as our eventual implementation. Besides, cross entropy loss is still employed in the implementation for a stable optimization.

**Experiments on Distribution Matching** In addition to methods constrained by gradient matching, the proposed robust DD can also be plugged into DD methods with other matching metrics. We evaluate the efficacy with IDM (Zhao et al., 2023) as the baseline, where distribution similarity is used as the matching metric. Similar to the integration in gradient matching, the CVaR loss is adopted during model training phases. As shown in the first two columns in Table 5, our proposed robust optimization achieves improvement across all metrics. The results demonstrate the possibility of applying RDD to broader dataset distillation methods for robustness enhancement.

**Scalability** We also scale up the proposed robust optimization to TinyImageNet, which contains 100 classes, and hence is more challenging compared with other benchmarks. The experiments are conducted on GLaD, and the results are shown in the last two columns of Table 5. When the class number to be optimized is increased, the proposed RDD method provides consistent improvement on the robustness of the distilled data by enhancing model training phases.

**Parameter Analysis** An analysis is conducted to evaluate the influence of different CVaR ratio $\alpha$ choices of equation 3 in Figure 2. We vary the $\alpha$ value from 0.2 to 1.0 to explore different ratios of data for calculating CVaR loss. Both the validation performance on the standard testing set and the worst sub-set accuracy (Cluster-min) reach their peak at $\alpha$=0.8. Including all the samples ($\alpha$=1.0) introduces certain interruptions for the optimization due to large loss values and results

in a slight performance drop. On the other hand, considering only a small portion of samples for CVaR loss loses essential information, leading to performance degradation. Notably, even the worst performance obtained at $\alpha$=0.2 is significantly higher than the GLaD baseline, particularly in terms of the Cluster-min metric. This observation strongly supports the effectiveness of our proposed robust dataset distillation method.

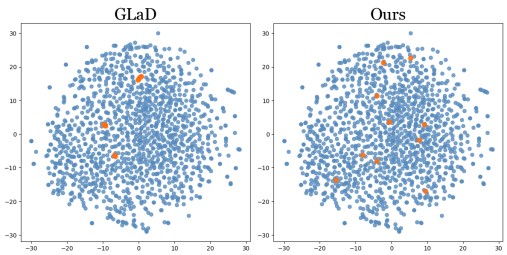 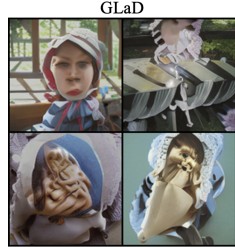 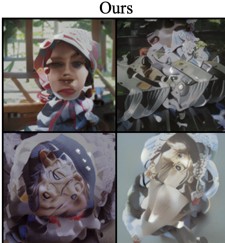

Figure 3: T-SNE distribution visualization of original samples (blue dots) and synthesized samples (orange dots) on ImageNet bonnet class.

Figure 4: Synthesized sample visualization comparison between GLaD and our proposed method. The samples are initialized identically.

**Visualization** To explicitly illustrate the impact of CVaR loss on the distillation results, we visualize the sample distribution comparison in Figure 3. In the feature space, samples optimized by GLaD tend to form small clusters, while the introduction of robust optimization leads to a more evenly distributed distilled dataset. Note that there is no constraint on the feature distribution applied during the distilling process. The proposed robust optimization involves loss calculation on different data partitions, contributing to better coverage over the original data distribution. The more even distribution observed further affirms the effectiveness of our proposed robust distillation method.

Additionally, we compare the synthesized samples of the same ImageNet bonnet class in the pixel space between the baseline GLaD and our proposed method in Figure 4. The images are initialized with the same original samples for better comparison. Remarkably, the additional CVaR loss introduces more irregular shapes into the image during optimization. These irregular shapes weaken specific features present in each image while introducing common features of the corresponding class, leading to a more even sample distribution in the latent space.

## 5    CONCLUSION

This paper explores the intricate relationship between DD and its generalization, with a particular focus on performance across uncommon subgroups, especially in regions with low population density. To address this, we introduce an algorithm inspired by distributionally robust optimization, employing clustering and risk minimization to enhance DD. Our theoretical framework, supported by empirical evidence, demonstrates the effectiveness, generalization, and robustness of our approach across diverse subgroups. By prioritizing representativeness and coverage over training error guarantees, the method offers a promising avenue for enhancing the models trained on synthetic datasets in real-world scenarios, paving the way for enhanced applications of DD in a variety of settings.

ACKNOWLEDGMENTS

The paper is supported by NSF 2112562, ARO W911NF-23-2-0224, and Czech National Science Foundation Project 24-11664S.

**Reproducibility Statement** We have provided detailed instructions to help reproduce the experimental results of this work. In Section. G we provide the statistics of the datasets used for evaluating the proposed method. In Section. H we provide implementation details on the baseline methods and the hyper-parameter settings of our method. The evaluation metric design of the domain-shift setting is also included. Additionally, we have attached the adopted source code in the supplementary material, which will further help understand the proposed method.

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

APPENDIX

The appendix is organized as follows: Sec. A provides more detailed theoretical analysis. Sec. C presents the proof for proposition 3.4. Sec. G and Sec. H contain the dataset details and implementation details, respectively. Sec. D offers more experimental results of the proposed robust dataset distillation method. Section I discusses the broader impact; and finally, Section F presents more sample visualization of the robust dataset distillation method.

# A    THEORETICAL DETAILS

## A.1    DRO AND GENERALIZATION

Here we present several results from the literature that indicate that a DRO on an empirical risk minimization problem exhibits generalization guarantees, i.e., approximately minimizes the population (or expected test dataset) loss. Consider equation 9, repeated here

$$\min_S \max_{\mathcal{Q} \in \mathbb{Q}} \max_{\mathcal{Q}' \in \bar{\mathcal{Q}}} \quad \mathbb{E}[F(\theta^*, \mathcal{Q}')]$$
$$\text{s.t.} \quad \theta^* \in \arg\min_\theta F(\theta, S)$$
$$\bar{\mathcal{Q}} = \{\mathcal{Q}' : I(\mathcal{Q}', \mathcal{S} \cup \mathcal{Q}_N) \leq r\}.$$

1. Theorem 3 in the work of Xu & Mannor (2012) provides a bound for the test error at the DRO solution.
2. Theorem 3.1 in the work of Zeng & Lam (2022) presents a probability that $|\mathbb{E}[F(\theta^*_{DRO}, \mathcal{Q})] - \mathbb{E}[F(\theta^*_{opt}, \mathcal{Q})]| \leq \epsilon$ as a function of $\epsilon$, where we denote the DRO and the exact minimum, respectively.

## A.2    LARGE DEVIATIONS AND DATA DRIVEN DRO

The analysis of the theoretical convergence and robustness properties of our method will rely significantly on the theoretical foundations of data driven DRO established in the work of Van Parys et al. (2021). To this end, we review a few pertinent definitions. A *predictor* is a function $c : \mathbb{R}^d \times \Xi \to \mathbb{R}$ that defines a model as applied to a data distribution. A *data driven predictor* uses an empirical distribution of samples $\hat{\mathbb{P}}_T$ in the prediction.

The *sample average predictor* is given as

$$c(\theta, \hat{\mathbb{P}}_T) = \frac{1}{T} \sum_{t=1}^T F(\theta, \xi_t)$$

Let $\hat{\theta} = \arg\min \hat{c}(\theta, \mathbb{P})$ be the data driven predictor.

An ordering $\preceq$ is introduced to rank the set of predictors, with

$$(\hat{c}_1, \hat{\theta}_1) \preceq (\hat{c}_2, \hat{\theta}_2) \text{ if and only if } \hat{c}_1(\hat{\theta}_1(\mathbb{P}'), \mathbb{P}') \leq \hat{c}_2(\hat{\theta}_2(\mathbb{P}'), \mathbb{P}'), \forall \theta, \mathbb{P}' \in \mathcal{P}$$

The *Distributionally Robust Predictor* is one defined as,

$$\hat{c}_r(\theta, \mathbb{P};) = \sup_{\mathbb{P} \in \mathcal{P}} \{c(\theta, \mathbb{P}) : \mathcal{D}_I(\mathbb{P}', \mathbb{P}) \leq r\}$$

where $\mathcal{D}_I(A, B)$ is the mutual information of random variables $A$ and $B$.

They define the optimization problem

$$\min_{(\hat{c}, \hat{\theta})} \quad (\hat{c}, \hat{\theta})$$
$$\text{s.t.} \quad \lim_{T \to \infty} \sum \frac{1}{T} \log \mathbb{P}^\infty \left( c(\hat{\theta}(\hat{\mathbb{P}}_T), \mathbb{P}) > \hat{c}(\hat{\theta}, \hat{\mathbb{P}}_T) \right) \leq -r, \forall \mathbb{P} \in \mathcal{P} \tag{15}$$

where the minimum is the lower bound with respect to the ordering $\preceq$ of all feasible (LDP satisfying) predictors.

Van Parys et al. (2021) prove that the distributionally robust predictor solves equation 15 in Theorem 4, for discrete distributions, and Theorem 6, for continuous ones.

We are constructing a synthetic dataset, while simultaneously considering training subsamples to form the empirical measure at each iteration, suggesting that the data-driven framework can fit the problem of interest. Formally, we consider solving optimization problems defined on a set of measures as decision variables,

$$\min_S \max_{\mathcal{Q} \in \mathbb{Q}} \quad \mathbb{E}[F(\theta^*, \mathcal{Q})]$$
$$\text{s.t.} \quad \theta^* \in \arg\min_\theta F(\theta, S)$$

by solving the data driven DRO approximation for the test error,

$$\min_S \max_{\mathcal{Q} \in \mathbb{Q}} \max_{\mathcal{Q}' \in \bar{\mathcal{Q}}} \quad \mathbb{E}[F(\theta^*, \mathcal{Q}')]$$
$$\text{s.t.} \quad \theta^* \in \arg\min_\theta F(\theta, S) \tag{16}$$
$$\bar{\mathcal{Q}} = \{\mathcal{Q}' : I(\mathcal{Q}', S \cup \mathcal{Q}_N) \leq r\}$$

Note that the form is the nested DRO problem described earlier. At the upper layer, there is an uncertainty set regarding the choice of $\mathcal{Q} \in \mathbb{Q}$, targeting group robustness. There is the data-driven DRO in the lower level, which is an algorithmic approximation of the population risk minimization with established accuracy guarantees.

## B   GENERALIZE TO BROADER DATASET DISTILLATION SCENARIOS

---

**Algorithm 2** Zero-order Dataset Distillation

---

**Input:** Initial synthetic set $S$, distilling objective $L(S)$

**Execute:**
    **For** epoch $E = 1, ...$
        Compute with $v_l \sim \mathcal{N}(0, I)$:

$$g(S) \approx \nabla L(S) = \frac{1}{M} \sum_{l=1}^{M} \frac{L(S + \sigma v_l) - L(S)}{\sigma} v_l \tag{17}$$

        Set stepsize $s$ (diminishing stepsize rule, or stepwise decay). For instance:

$$s = 0.1/\sqrt{1 + E} \tag{18}$$

        Set $S - sg \rightarrow S$

---

Existing dataset distillation methods adopt differentiable matching metrics to optimize the synthetic data $S$. However, there can be circumstances where the gradient of the objective cannot be directly acquired. In this case, dataset distillation can still be performed by zero order approximation of the gradient. Starting with some initial $S$ and proceed as in Algorithm 2, a standard gradient descent approach can be applied, using a zero order approximation of the gradient of $L$ concerning the dataset $S$, that is $g(S) \approx \nabla L(S)$, computed only using evaluations of $L(S)$.

We can use any method from Berahas et al. (2022) to obtain a gradient estimate of $g \approx \nabla_S L(S)$, here the featured procedure is Gaussian smoothing. The proposed robust dataset distillation method is still able to be applied towards $L(S)$ for more robust optimization.

## C   PROOF OF PROPOSITION 3.4

Consider the quantity:

$$\frac{1}{N} \log \mathbb{P}_{\mathcal{Q}}^\infty \left( F(\theta^*(S), \mathbb{P}_\mathcal{Q}) > F(\theta^*(S), S \cup \mathcal{Q}_N) \right) \tag{19}$$

Now recall the expression for CVaR:

Table 6: Top-1 test accuracy on CUB-200 transferred from ImageNet subsets. All the results are reported based on the average of 5 runs. $^{\mathcal{R}}$ indicates the proposed robust dataset distillation method applied. The better results between baseline and the proposed method are marked in **bold**.

| Transfer Learning | GLaD | GLaD$^{\mathcal{R}}$ |
|---|---|---|
| ImageNet-50 $\rightarrow$ CUB-200 | $49.2_{\pm 0.9}$ | $\mathbf{54.6}_{\pm 0.6}$ |

Table 7: Top-1 test accuracy on more challenging test on the syn data trained model. All the results are reported based on the average of 5 runs. $^{\mathcal{R}}$ indicates the proposed robust dataset distillation method applied. The better results between baseline and the proposed method are marked in **bold**.

| Group | GLaD | GLaD$^{\mathcal{R}}$ |
|---|---|---|
| ImageNet-10 $\rightarrow$ ImageNet-A | $56.0_{\pm 1.1}$ | $\mathbf{59.6}_{\pm 0.9}$ |
| ImageNet-10 $\rightarrow$ ImageNet-B | $43.0_{\pm 0.9}$ | $\mathbf{47.2}_{\pm 1.0}$ |

$$\texttt{CVaR}[F(\theta; c_i)] = -\frac{1}{\alpha} \left\{ \frac{1}{|c_i|} \sum_{z_t \in c_i} F(\theta; z_t) 1[F(\theta; z_t) \leq f_\alpha] \right.$$
$$\left. + f_\alpha(\alpha - \frac{1}{|c_i|} \sum_{z_t \in c_i} 1[F(\theta; z_t) \leq f_\alpha]) \right\}$$

and

$$f_\alpha = \min\{f \in \mathbb{R} : \frac{1}{|c_i|} \sum_{z_t \in c_i} 1[F(\theta; z_t) \leq f_\alpha] \geq \alpha$$

**Step 1:** CVaR itself satisfies a LDP (Brown, 2007; Mhammedi et al., 2020). With these results, we can relate the empirical to the population CVaR.

**Step 2:** Notice that as long as $\alpha < \frac{1}{2}$, then minimizing the CVaR will also minimize equation 19.

**Step 3:** There is at least one estimator satisfying the LDP, namely, the DRO. Thus minimizing the empirical CVaR, by the feasibility of the LDP, must result in an estimator that satisfies the LDP.

# D  MORE EXPERIMENTS AND ANALYSIS

## D.1  MORE CHALLENGING TESTING SCENARIOS

In Section 4, we demonstrate that the proposed robust dataset distillation method is able to enhance the robustness of the generated data on shifted or truncated testing sets of the same classes. In this section, we evaluate RDD on more challenging testing cases.

**Downstream Fine-tuning**  The robustness of a neural network can be tested in transfer learning scenarios (Djolonga et al., 2021). Accordingly, we conduct the experiment of transferring the model trained on distilled data to downstream tasks. More concretely, the model is first pre-trained on distilled data from the 50-class subset of ImageNet (Cazenavette et al., 2023). Then it is fine-tuned on CUB-200 Wah et al. (2011). We use the top-1 accuracy on CUB-200 to evaluate the transferability of the distilled data, and the results are shown in Table 6. The results suggest that the data distilled with RDD applied significantly enhances the transferability of the trained model.

**One-shot Direct Transfer**  A more challenging case would be directly applying the model trained on the distilled data on completely different classes. As in the original model, a classifier is involved to predict the classification probability, directly applying the classifier to different classes would be infeasible. Thus, we adopt a one-shot retrieval-style evaluation approach instead. In this approach, the model is initially trained using the distilled data of 10 classes and subsequently tested on another set of 10 classes. During testing, the corresponding images are passed through the backbone to obtain embedded features, with which a similarity matrix is computed. We use each sample as a query, and check whether the most similar sample among the remaining samples belongs to the same class and report the top-1 accuracy.

Table 8: Top-1 test accuracy on standard and robustness testing sets. All the results are reported based on the average of 5 runs. The experiment is conducted with DREAM and PDD on the CIFAR-10 IPC-50 setting. $\mathcal{R}$ indicates the proposed robust dataset distillation method applied. The better results between baseline and the proposed method are marked in **bold**.

| Method | Acc | Cluster-min | Noise | Blur | Invert |
|---|---|---|---|---|---|
| PDD | $67.9_{\pm 0.2}$ | $63.9_{\pm 0.4}$ | $58.2_{\pm 0.7}$ | $48.9_{\pm 1.1}$ | $25.7_{\pm 0.5}$ |
| PDD$^{\mathcal{R}}$ | $\mathbf{68.7}_{\pm 0.6}$ | $\mathbf{65.1}_{\pm 0.3}$ | $\mathbf{59.4}_{\pm 0.9}$ | $\mathbf{50.6}_{\pm 0.9}$ | $\mathbf{26.7}_{\pm 0.7}$ |
| DREAM | $69.4_{\pm 0.4}$ | $64.7_{\pm 0.6}$ | $58.8_{\pm 1.2}$ | $50.1_{\pm 0.8}$ | $25.7_{\pm 0.6}$ |
| DREAM$^{\mathcal{R}}$ | $\mathbf{69.7}_{\pm 0.5}$ | $\mathbf{65.5}_{\pm 0.4}$ | $\mathbf{59.8}_{\pm 0.9}$ | $\mathbf{50.7}_{\pm 0.7}$ | $\mathbf{26.9}_{\pm 0.8}$ |

Table 9: Top-1 test accuracy on different cluster numbers. All the results are reported based on the average of 5 runs. The experiment is conducted with GLaD on CIFAR-10 under the IPC of 50. $\mathcal{R}$ indicates the proposed robust dataset distillation method applied. The best results are marked in **bold**.

| Cluster | Acc | Cluster-min | Noise | Blur | Invert |
|---|---|---|---|---|---|
| 5 | $61.6_{\pm 0.8}$ | $55.1_{\pm 0.9}$ | $54.6_{\pm 0.7}$ | $40.3_{\pm 0.5}$ | $17.5_{\pm 0.9}$ |
| 10 | $\mathbf{62.5}_{\pm 0.7}$ | $\mathbf{56.6}_{\pm 1.0}$ | $\mathbf{55.7}_{\pm 0.7}$ | $\mathbf{41.1}_{\pm 0.6}$ | $\mathbf{18.0}_{\pm 0.9}$ |
| 20 | $60.9_{\pm 0.6}$ | $54.3_{\pm 1.2}$ | $52.8_{\pm 0.4}$ | $39.8_{\pm 0.5}$ | $17.3_{\pm 0.7}$ |
| 30 | $59.9_{\pm 0.7}$ | $53.2_{\pm 0.9}$ | $51.5_{\pm 0.6}$ | $38.9_{\pm 0.5}$ | $17.1_{\pm 1.0}$ |

The results are presented in Table 7. For these two groups, the model is firstly trained on distilled ImageNet-10 and then tested on ImageNet-A and ImageNet-B, respectively. The subset split of ImageNet-A and ImageNet-B follows the setting in GLaD (Cazenavette et al., 2023). With the robust optimization applied, the distilled data also shows better robustness on unseen classes, demonstrating the efficacy of the proposed method in enhancing the generalizability of the distilled data.

### D.2 APPLICATION ON MORE DD METHODS

In addition to IDC (Kim et al., 2022) and GLaD (Cazenavette et al., 2023), we further conduct experiments on CIFAR-10 with DREAM (Liu et al., 2023) and PDD (Chen et al., 2024) as baselines in Table 8. Similar to the implementation for IDC, we use the extra CVaR criterion during the model updating. By the application of robust optimization, all the reported metrics have been improved, especially on the domain-shifted settings. This result demonstrates the generality of the proposed framework across DD methods.

### D.3 EFFICIENCY EVALUATION

As the proposed robust dataset distillation method involves extra CVaR loss calculation during the model updating, the efficiency issue might be a concern. Accordingly, we record the required time to complete the extra robust optimization on IDC. The baseline requires 70s to finish a loop, while the CVaR loss calculation takes up extra 30s. The robust optimization takes up less than 50% of the original calculation time. The extra time is not overwhelming, but brings significant improvements on the robustness of the distilled data.

### D.4 CHOICE OF THE NUMBER OF CLUSTERS

The CVaR loss calculation involves separating the training set into several sub-sets. Different settings of sub-set division can have influence on the CVaR optimization effects. In this work, the choice of the number of clusters is primarily based on common settings observed in DD benchmarks. Typically, our method is evaluated across different Images-per-class (IPC) settings, such as 1, 10, and 50. For an IPC below 10, we simply use the IPC as the number of clusters, and treat the synthetic samples as the cluster center. However, when dealing with a larger IPC, too many clusters might potentially result in insufficient samples in each cluster for CVaR loss calculation. To address this concern, we conduct a parameter analysis on the number of clusters specifically under an IPC of 50, where the number of clusters varies from 5 to 30. Synthetic samples of the same number are randomly selected to serve

Table 10: Top-1 test accuracy on different mini-batch sizes. All the results are reported based on the average of 5 runs. The experiment is conducted with GLaD on CIFAR-10 under the IPC of 10. $\mathcal{R}$ indicates the proposed robust dataset distillation method applied. The best results are marked in **bold**.

| Mini-batch Size | Acc | Cluster-min |
|---|---|---|
| 64 | $46.9_{\pm0.7}$ | $40.3_{\pm1.1}$ |
| 128 | $48.9_{\pm0.8}$ | $45.1_{\pm0.8}$ |
| 256 | $\mathbf{50.2}_{\pm0.5}$ | $\mathbf{46.7}_{\pm0.9}$ |
| 512 | $49.3_{\pm0.5}$ | $45.7_{\pm1.2}$ |

Table 11: Ablation study on the initialization of synthetic samples. All the results are reported based on the average of 5 runs. $\mathcal{R}$ indicates the proposed robust dataset distillation method applied. "Cluster" means that the synthetic samples are initialized with clustering centers.

| Dataset | IPC | GLaD | GLaD$^{\mathcal{R}}$ | GLaD$^{\mathcal{R}}$+Cluster |
|---|---|---|---|---|
| CIFAR-10 | 1 | $28.0_{\pm0.8}$ | $29.2_{\pm0.8}$ | $29.4_{\pm0.9}$ |
| | 10 | $46.7_{\pm0.6}$ | $50.2_{\pm0.5}$ | $50.3_{\pm0.6}$ |
| ImageNet-10 | 1 | $33.5_{\pm0.9}$ | $36.4_{\pm0.8}$ | $36.5_{\pm0.8}$ |
| | 10 | $50.9_{\pm1.0}$ | $55.2_{\pm1.1}$ | $55.0_{\pm1.2}$ |

as cluster centers, and sub-sets are accordingly separated based on these centers. The experiment is conducted with GLaD on CIFAR-10, and the results are shown in Table 9.

When the number of clusters increases over IPC, as the total training sample number is fixed, the sample number belonging to each sub-set would decrease. And the insufficient samples from each cluster cause sub-optimal CVaR optimization, leading to a performance drop. By empirical observation, we fix the number of clusters to 10 for large-IPC settings.

### D.5 Choice of Mini-batch Size

As the group DRO does require abundant samples for precise calculation, we set the sample number in a mini-batch as 256 in our experiments. We further conduct an analysis on the parameter setting, and the results are listed in Table 10 (on CIFAR-10). When there are limited samples in a mini-batch, the performance will be largely influenced. Applying a mini-batch size around 256 brings a mild influence on the validation performance. The slight performance drop when the mini-batch size is set as 512 might be due to less diversity between different mini-batches.

### D.6 Choice of Initialization

As the clusters are separated based on the synthetic samples, different initialization of synthetic samples can influence the effects of CVaR loss calculation. In our experiments, we follow the baseline setting, which is random initialization with real samples for both IDC (Kim et al., 2022) and GLaD (Cazenavette et al., 2023). We accordingly conduct an ablation study to evaluate the robustness of RDD on the initialization. In addition to random sampling, we also test initializing the synthetic samples with clustering centers, which leads to a more even distribution. The results are shown in Table 11, where clustering initialization yields similar results to random initialization. Although random initialization cannot guarantee an even distribution at the beginning, during the subsequent optimization process, the algorithm is still robust enough to handle different initializations and provide stable performance improvement over the baseline.

### D.7 Evaluation on Domain Generalization Baselines

The proposed robust dataset distillation method mainly focuses on enhancing the robustness of distilled datasets. The method can also be combined with other domain generalization methods to further enhance the robustness of model training. We have included the three generalization methods MMD (Li et al., 2018), RSC (Huang et al., 2020), and HYPO (Bai et al., 2024) in Table 12 to serve as training pipelines. The results suggest that on the one hand, RDD consistently provides performance

Table 12: Evaluation of the proposed robust dataset distillation on other domain generalization methods with GLaD. All the results are reported based on the average of 5 runs. $\mathcal{R}$ indicates the proposed robust dataset distillation method applied. The better results between baseline and the proposed method are marked in **bold**.

| Method | CIFAR-10 | | ImageNet-10 | |
|---|---|---|---|---|
| | GLaD | GLaD$^{\mathcal{R}}$ | GLaD | GLaD$^{\mathcal{R}}$ |
| Baseline | $46.7_{\pm 0.6}$ | $\mathbf{50.2}_{\pm 0.5}$ | $50.9_{\pm 1.0}$ | $\mathbf{55.2}_{\pm 1.1}$ |
| MMD | $47.1_{\pm 0.6}$ | $\mathbf{51.3}_{\pm 0.5}$ | $51.8_{\pm 1.3}$ | $\mathbf{56.5}_{\pm 1.2}$ |
| RSC | $47.9_{\pm 0.5}$ | $\mathbf{52.5}_{\pm 0.5}$ | $52.4_{\pm 1.1}$ | $\mathbf{56.8}_{\pm 1.0}$ |
| HYPO | $49.0_{\pm 0.5}$ | $\mathbf{53.2}_{\pm 0.6}$ | $53.6_{\pm 1.2}$ | $\mathbf{58.1}_{\pm 1.1}$ |

Table 13: Validation accuracy on more ImageNet sub-sets in comparison with GLaD. The data is distilled with ConvNet-5 and evaluated with ResNet-10 under the IPC of 10. $\mathcal{R}$ indicates the proposed robust dataset distillation method applied. The better results between baseline and the proposed method are marked in **bold**.

| Dataset | Acc | | Cluster-min | |
|---|---|---|---|---|
| | GLaD | GLaD$^{\mathcal{R}}$ | GLaD | GLaD$^{\mathcal{R}}$ |
| ImageNet-A | $53.9_{\pm 0.7}$ | $\mathbf{57.5}_{\pm 1.2}$ | $40.5_{\pm 0.9}$ | $\mathbf{45.8}_{\pm 0.8}$ |
| ImageNet-B | $50.3_{\pm 0.9}$ | $\mathbf{53.8}_{\pm 0.8}$ | $42.9_{\pm 1.1}$ | $\mathbf{47.0}_{\pm 1.0}$ |
| ImageNet-C | $49.2_{\pm 0.8}$ | $\mathbf{51.3}_{\pm 0.6}$ | $28.2_{\pm 0.7}$ | $\mathbf{32.8}_{\pm 0.6}$ |
| ImageNet-D | $39.1_{\pm 0.6}$ | $\mathbf{40.9}_{\pm 0.7}$ | $27.0_{\pm 0.8}$ | $\mathbf{31.3}_{\pm 0.9}$ |
| ImageNet-E | $38.9_{\pm 0.8}$ | $\mathbf{41.1}_{\pm 0.9}$ | $25.8_{\pm 0.6}$ | $\mathbf{30.0}_{\pm 0.7}$ |

improvement on different training pipelines. On the other hand, the combination of RDD and other domain generalization methods can further improve the results over the baseline.

## D.8 RESULTS ON MORE IMAGENET SUB-SETS

GLaD (Cazenavette et al., 2023) designs experiments on multiple ImageNet sub-sets. We further provide results on these sub-sets in comparison with GLaD in Table 13. The sub-set division remains consistent with that in the work of Cazenavette et al. (2023). Our proposed robust DD method consistently outperforms the baseline, particularly in terms of the worst accuracy across clustered testing sub-sets. This observation underscores the stability and versatility of the proposed method.

## E RELATED WORK

**Dataset Distillation** Dataset distillation (DD) seeks to distill the richness of extensive datasets into compact sets of synthetic images that closely mimic training performance (Sachdeva & McAuley, 2023). These condensed images prove invaluable for various tasks, including continual learning (Gu et al., 2023b), federated learning (Liu et al., 2022a; Jia et al., 2023), neural architecture search (Such et al., 2020; Medvedev & D'yakonov, 2021), and semi-supervised learning (Vahidian et al., 2020; Joneidi et al., 2020). Existing DD methodologies can be broadly categorized into bi-level optimization and training metric matching approaches. Bi-level optimization integrates meta-learning into the surrogate image update process with the validation performance as a direct optimizatino target (Zhou et al., 2022; Loo et al., 2023). Conversely, metric matching techniques refine synthetic images by aligning with training gradients (Kim et al., 2022; Liu et al., 2023), feature distribution (Sajedi et al., 2023; Zhao et al., 2023), or training trajectories (Wu et al., 2023; Du et al., 2023) compared to the original images. Data parametrization (Kim et al., 2022; Liu et al., 2022b; Wei et al., 2024) and generative prior (Cazenavette et al., 2023; Gu et al., 2023a; Wang et al., 2023a) are also considered for more efficient DD method construction.

**Robustness in Dataset Distillation** To the best of our knowledge, this is the first work on subgroup accuracy specifically, or even DRO generally, for Dataset Distillation. We consider a few classes of related works. In the context of DD, adversarial robustness is another popular notion of robustness,

Table 14: The configurations of different datasets

| Training Dataset | Class | Images per class | Resolution |
|---|---|---|---|
| SVHN | 10 | $\sim 6000$ | 32×32 |
| CIFAR10 | 10 | 5000 | 32×32 |
| ImageNet-10 | 10 | $\sim 1200$ | 128×128 |
| ImageNet subsets | 10 | $\sim 1200$ | 128×128 |

which is simply minimization with respect to the worst possible sample in the support of some perturbation on the data sample, rather than the distribution. Adversarial Robustness is more conservative than DRO, however, in some security-sensitive circumstances, this is a more appropriate framework (Wu et al., 2022; Xue et al., 2024). In the field of knowledge distillation, wherein a more parsimonious model, rather than dataset, is of interest, group DRO has been considered in the work of Vilouras et al. (2023) and Wang et al. (2023b).

The concept of group DRO is considered and an implementation thereof is demonstrated in the work of Sagawa et al. (2019), complemented with a regularization strategy that appears to assist in performance for small population subgroups. The coverage of disparate data distributions is also a concern in Non-IID federated learning. Jiao et al. (2023) present convergence theory for DRO in a federated setting.

## F  VISUALIZATION OF SYNTHESIZED SAMPLES

We provide the visualization of synthesized samples in different datasets in Figure 5 to Figure 8. Each row represents a class.

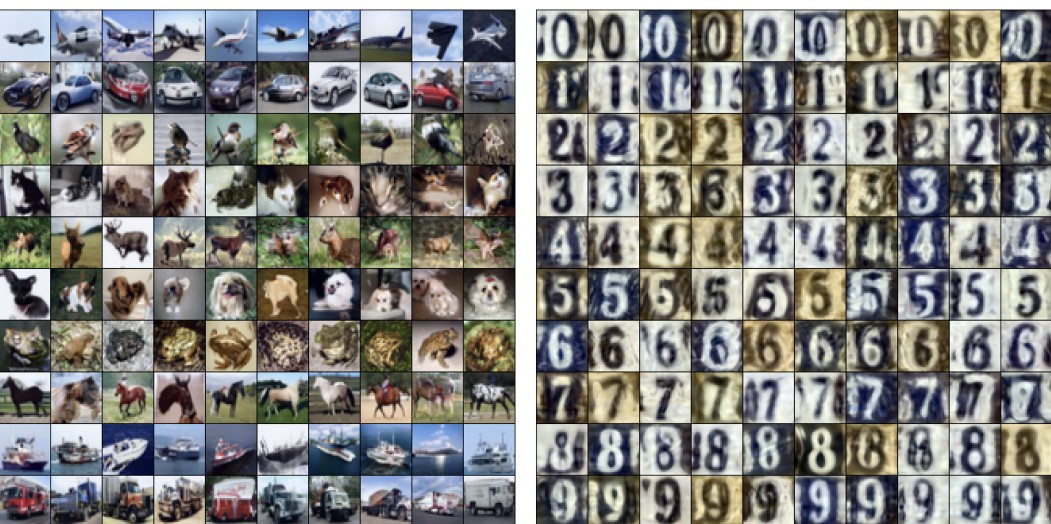

Figure 5: Synthesized samples of CIFAR-10 (left) and SVHN (right).

## G  DATASET STATISTICS

We evaluate our method on the following datasets:

- **SVHN** (Yuval, 2011) is a dataset for digits recognition cropped from pictures of house number plates that is widely used for validating image recognition models. It includes 600,000 32×32 RGB images of printed digits ranging from 0 to 9. SVHN comprises three subsets: a training set, a testing set, and an extra set of 530,000 less challenging images that can aid in the training process. SVHN dataset is released with a CC0:Public Domain license.

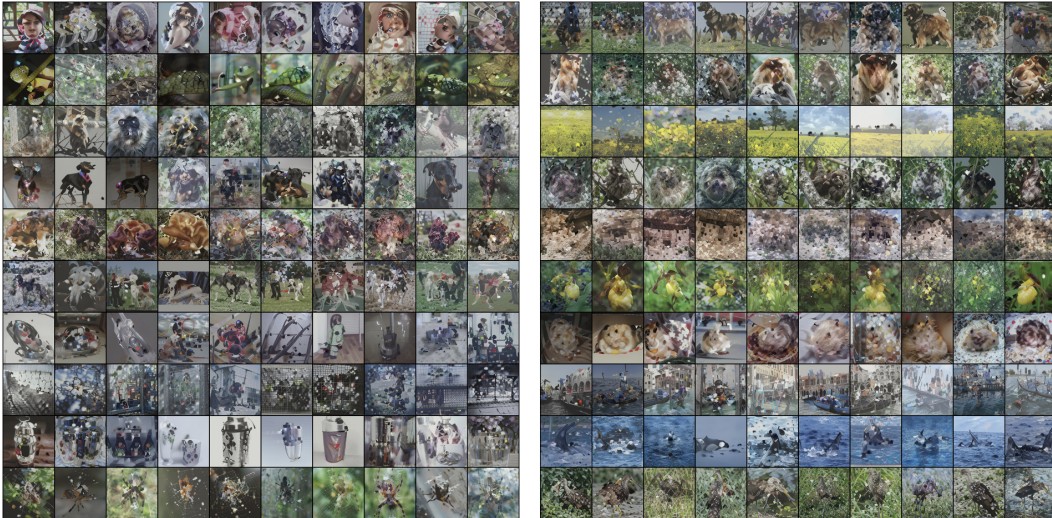

Figure 6: Synthesized samples of ImageNet-10 (left) and ImageNet-A (right).

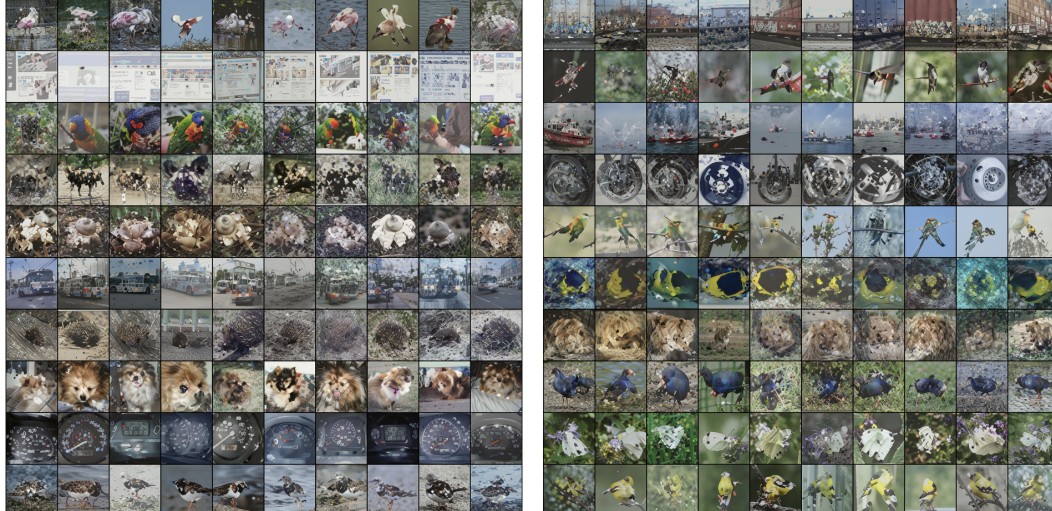

Figure 7: Synthesized samples of ImageNet-B (left) and ImageNet-C (right).

- **CIFAR-10** (Krizhevsky et al., 2009) is a subset of the Tiny Images dataset and consists of 60000 32x32 color images. The images are labeled with one of 10 mutually exclusive classes: airplane, automobile, bird, cat, deer, dog, frog, horse, ship, and truck. There are 6000 images per class with 5000 training and 1000 testing images per class. CIFAR-10 dataset is released with an MIT license.
- **ImageNet-10** and **ImageNet subsets** is the subset of ImageNet-1K (Deng et al., 2009) containing 10 classes, where each class has approximately 1, 200 images with a resolution of $128 \times 128$. The individual configurations of these datasets are shown in Table 14. No license is specified for ImageNet.

## H IMPLEMENTATION DETAILS

The proposed method can be applied to various popular dataset distillation frameworks. In this paper, we perform experiments on pixel-level distillation method IDC (Kim et al., 2022) and latent-level method GLaD (Cazenavette et al., 2023) to substantiate the consistent efficacy of our method. The experiments are conducted on popular dataset distillation benchmarks, namely SVHN, CIFAR-10,

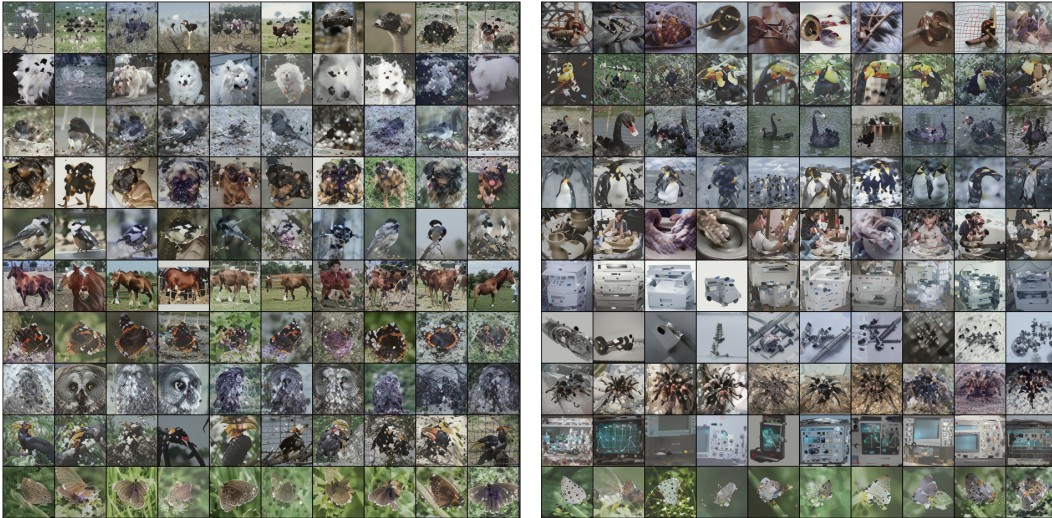

Figure 8: Synthesized samples of ImageNet-D (left) and ImageNet-E (right).

and ImageNet (Yuval, 2011; Krizhevsky et al., 2009; Deng et al., 2009). The images of SVHN and CIFAR-10 are resized to $32 \times 32$, while those from ImageNet-10 are resized to $128 \times 128$, representing diverse resolution scenarios. The split of ImageNet-10 subset follows Kim et al. (2022)

The CVaR loss and the Cluster-min metric calculation both involve clustering. For the CVaR loss, Euclidean distance is adopted to evaluate the sample relationships. The real samples are assigned to the synthetic sample with the smallest distance. Due to the CVaR loss calculation involving an ample number of samples, the mini-batch size during model updating is increased to 256. In cases where the IPC setting is less than 10, the cluster number in Eq. 2 is set equal to IPC. For larger IPCs, the cluster number is fixed at 10, with 10 random synthesized samples chosen as the clustering centers. The ratio $\alpha$ in CVaR loss is set to 0.8. For the Cluster-min metric calculation, we first apply the standard KMeans algorithm to partition the original test set into 10 subsets. Specifically, as outlined in Algorithm 1, the subsampling and clustering processes are performed within each class. For the cluster-min metric calculation, we execute the clustering in the RGB space. This method ensures that clusters are formed based on common features shared by these samples rather than random selection, resulting in clusters that include samples from different classes. We have also verified that each cluster contains samples from all classes.

For the Cluster-min metric calculation, a standard KMeans algorithm is conducted to separate the original testing set into 10 sub-sets. To clarify, According to Algorithm 1, the subsampling and clustering processes are based on each class. As for the cluster-min metric calculation, we conduct the clustering in the RGB space. This approach ensures that clusters are formed based on common features rather than random selection and include samples from different classes. We have examined the cluster division to make sure that each cluster contains samples from all classes.

All the experiments are conducted on a single 24G RTX 4090 GPU.

For a fair comparison, the experiment settings are generally kept the same as in the original papers. Detailed explanations are listed below for both baselines.

## H.1   IDC DETAILS

A multi-formation operation is proposed in IDC to increase the information contained in each sample. The multi-formation factor is set as 2 on CIFAR-10 and 3 on ImageNet, which is kept the same as the original implementation.

On SVHN and CIFAR-10 datasets, a 3-layer ConvNet (Gidaris & Komodakis, 2018) is employed for distillation, while on ImageNet, ResNet-10 (He et al., 2016) is utilized. After distillation, we conduct

validation procedures on ConvNet-3 for 32×32 datasets and ResNet-10 for ImageNet, ensuring a fair and consistent basis for comparison.

## H.2 GLaD DETAILS

There are multiple different matching metrics presented in GLaD. Here we adopt gradient matching (DC in GLaD paper) in our experiments for two primary reasons. Firstly, gradient matching is deemed to be more practical when compared with alternative metrics. Secondly, gradient matching incorporates model updating, providing a convenient avenue for embedding the proposed distributionally robust optimization method.

On SVHN and CIFAR-10 datasets, similar to IDC, a 3-layer ConvNet (Gidaris & Komodakis, 2018) is employed for distillation. A 5-layer ConvNet is adopted for ImageNet, which is kept the same as in the original paper. For better comparison, we adopt the validation protocol in IDC as it yields better performance. The results of baseline GLaD are also re-produced with the same protocol.

## H.3 DISTILLING ITERATION

Initially, the iteration randomly initializes a network according to the architecture setting for each baseline. Then the synthetic images are updated to match the gradients from the network. The network supplying the gradient for the images is updated following the image update. During the network updates, the robust objective proposed in this paper is employed for training. The network training is restricted to an early stage, using only 4,000 images for IDC and 1,000 images for GLaD. 100 steps of synthetic image update together with the network training forms an iteration for IDC, and 2 steps for GLaD. For CIFAR-10 and ImageNet subsets, we adopt 2,000 and 500 iterations to complete the distilling process, respectively.

# I    BROADER IMPACTS

The primary objective of dataset distillation is to alleviate the storage and computational resource demands associated with training deep neural networks. This need becomes particularly pronounced in the era of foundational models. Dataset distillation endeavors to expedite environmental sustainability efforts. Our proposed method, viewed from this perspective, markedly diminishes the resources needed for the distillation process. We aspire to draw attention to practical dataset distillation methods within the computer vision community, thereby fostering the sustainable development of society. Further, this work does not involve direct ethical concerns. Our experiments utilize publicly available datasets, namely ImageNet, SVHN, and CIFAR-10. In forthcoming research, we are committed to addressing issues related to generation bias and diversity when constructing small surrogate datasets.

