# OpenReview forum: "Group Distributionally Robust Dataset Distillation with Risk Minimization"
_ICLR.cc/2025/Conference — ICLR 2025 Poster_

### Official Review · Reviewer_gBwM · 2024-11-02

**Soundness:** 3
**Presentation:** 3
**Contribution:** 3
**Rating:** 6
**Confidence:** 4

**Summary:**

This paper proposes a robust dataset distillation method that enhances generalization across underrepresented subgroups by using a double-layer distributionally robust optimization (DRO) approach. Traditional dataset distillation often compresses training data into synthetic sets by matching training characteristics but may struggle with subgroup generalization. To improve this, the authors cluster the training data around synthetic data points and apply a Conditional Value at Risk (CVaR) loss to minimize worst-case subgroup errors, making the synthetic dataset more representative of diverse data distributions. Experimental results show that this method significantly improves robustness under domain shifts, outperforming baseline methods on benchmarks like CIFAR-10 and ImageNet-10, particularly in challenging conditions like noise and blurring.

**Strengths:**

- As far as I know, this paper firstly addresses a major limitation in standard dataset distillation (underrepresented or rare subgroups) by applying a double-layer distributionally robust optimization (DRO) framework. This approach ensures the synthetic dataset better represents the full diversity of the data, reducing performance drops across different data subgroups.

- This paper provides not only empirical evidence of the proposed method's robustness against domain shifts such as noise, blur, and inversion, but also a theoretical analysis of the effectiveness of CVaR in the loss function to enhance robustness.

- The proposed method is designed to be adaptable and can integrate with various existing distillation techniques, such as gradient or distribution matching. This modularity makes it compatible with a range of distillation methods, which are still actively developing and evolving.

**Weaknesses:**

- The proposed DRO approach, particularly with clustering and CVaR, may introduce significant computational overhead. This added complexity could be amplified for large-scale datasets due to the clustering of training (real) data. However, the paper does not discuss the computational overhead of the proposed method.

- Although the method shows promising results on benchmarks like CIFAR-10 and ImageNet-10, the experiments are limited to controlled domain shifts (e.g., noise, blur). Testing under more realistic settings, such as in transfer learning, would further validate its robustness and practical relevance. For example, one could (1) train neural networks on a synthetic dataset distilled from a coarse-grained dataset (e.g., ImageNet) and (2) fine-tune and evaluate them on a fine-grained dataset (e.g., Birds 200). This setup would better illustrate the method's effectiveness in addressing the challenges posed by rare subgroups.

**Questions:**

Please see weaknesses section.

---

> ### Author Response · Authors · 2024-11-21
> **Response to Reviewer gBwM**
>
> Thank you for your detailed and constructive comments. Please find the responses below:
>
> ### **1. Computational overhead analysis [W1]**
> Thanks for your constructive advice.
> Indeed the introduction of clustering and CVaR adds extra computation time to the algorithm. We have included an efficiency evaluation in **Section D.3** of the appendix. The baseline requires 70 seconds to finish a loop, while the CVaR loss calculation takes up an extra 30 seconds.
> The robust optimization takes up less than 50% of the original calculation time.
> The extra time is within an acceptable range, while providing significant improvement on the robustness.
> We will make a clearer presentation in the revision. We will also aim to optimize the clustering implementation to further reduce the extra computational cost.
>
> We would also want to clarify that using CVaR instead of a sample average only takes a constant multiple of operations on the subsample. Clustering itself can be done polynomial in the number of samples. Since the properties of the model and loss function, that is, nonconvex and nonsmooth, are fundamentally unchanged, there is no change to the iteration complexity. We have added this part to **line 232** of the revised manuscript.
>
> ### **2. Experiment on transfer learning [W2]**
>
> This is an insightful opinion. We accordingly perform the following experiment:
> - Train a model with data distilled from 50 ImageNet classes (ImageNet-A to ImageNet-E).
> - Transfer the model to CUB-200.
>
> The results of baseline and RDD are presented in the following table and **Table 5** of the revised manuscript:
>
> | Transfer Learning | GLaD | GLaD+RDD |
> | --- | --- | --- |
> | ImageNet $\rightarrow$ CUB-200 | 49.2 $\pm$ 0.9 | **54.6** $\pm$ 0.6 |
>
> The results show that RDD also enhances the generalization performance in this case.
>
> In addition, we also present another transfer experiment in **Section D.1** of the appendix. We first train the model with data distilled in ImageNet-10. Then, without fine-tuning, the model is directly applied to two other un-overlapped ImageNet subsets.
> The results are obtained by evaluating if the query sample and the most similar sample belong to the same class (similar to a 1-shot learning setting).
> And our proposed RDD illustrates substantial improvement in the transfer performance.

---

### Official Review · Reviewer_1jHv · 2024-11-03

**Soundness:** 3
**Presentation:** 3
**Contribution:** 3
**Rating:** 8
**Confidence:** 4

**Summary:**

This paper proposes a new data distillation approach emphasizing the subgroup coverage and generalization ability of the models trained on the synthetic dataset. This paper provides a theoretical analysis rooted in Distributionally Robust Optimization (DRO) and verifies the effectiveness of the proposed method with various experiments.

**Strengths:**

* This paper considers the group coverage and generalization ability of the synthetic dataset in Data Distillation(DD), which is interesting and novel.

* The introduced algorithm is clear and the theoretical analysis seems solid.

* The numerical results do show the effectiveness of the proposed algorithm. Meanwhile, the figures (Fig.3, 4) seem to indicate that the proposed method improves group coverage.

**Weaknesses:**

* While the paper emphasizes group coverage and generalization in data distillation, the experiments are mainly conducted on IID datasets. More experimental results in scenarios with distribution shift between training and testing sets (such as long-tail classification, subpopulation shift, domain adaptation, domain generalization, etc) can further validate the improvement in group coverage and generalization ability,

* According to Algorithm 1, the initialization of the synthetic dataset seems very important because it involves how the training data samples are clustered into subgroups. It may require further ablation study to verify the stability of the proposed algorithm.

* The proposed method seems like a plug-in as it only modifies the objective for data distillation and could be combined with any data distillation methods. A more sophisticated comparison between the proposed method with other objectives regarding training gradients, feature distributions, and training trajectory could help readers better understand the improvement of the proposed method.

**Questions:**

I have stated many of my suggestions and concerns in the weaknesses section. Below, I have further questions with some minor issues.

* While theoretical analyses have been provided, I wonder whether the proposed method would affect the convergence rate and make it more difficult to find an optimal solution for the proposed objective. I would appreciate an empirical time complexity analysis regarding the proposed method.

* In Eq.10, what does the *N* represent? I did not see any introduction of the *N*.

---

> ### Author Response · Authors · 2024-11-21
> **Response to Reviewer 1jHv (1/2)**
>
> Thank you for your detailed and constructive comments. Please find the responses below:
>
> ### **1. Experiments on scenarios with distribution shifts [W1]**
> Thanks for the constructive question. As suggested, we have conducted an extended experiment in domain transfer and subpopulation shift settings.
>
> - In the first domain transfer experiment, the model is first trained with data distilled from a 50-class ImageNet subset (ImageNet-A - ImageNet-E in the paper).
> Then it is fine-tuned on the fine-grained CUB-200 dataset, and the evaluation top-1 accuracy is shown in the following table. The results suggest that our proposed robust dataset distillation can also substantially improve the transfer performance on downstream tasks.
>
> | Transfer Learning | GLaD | GLaD+RDD |
> | --- | --- | --- |
> | ImageNet-50 $\rightarrow$ CUB-200 | 49.2 $\pm$ 0.9 | **54.6** $\pm$ 0.6 |
>
> - In the second subpopulation shift experiment, the model is trained on the data distilled from the MetaShift benchmark, which involves spurious correlations. The results are shown below:
>
> | Metric | GLaD | GLaD+RDD |
> | --- | --- | --- |
> | Average Accuracy | 58.6 $\pm$ 2.3 | **62.2** $\pm$ 1.2 |
> | Worst-group Accuracy | 51.3 $\pm$ 1.8 ($\downarrow$ 7.3) | **57.0** $\pm$ 1.0 ($\downarrow$ 5.2) |
>
> In addition, we have included the domain generalization experiment in **Section D.1** of the appendix.
> The model is trained on one 10-class subset of ImageNet and tested on another 10-class subset.
> As the linear classifier is not trained on the target subset, we conduct a one-shot retrieval style evaluation to check if the most similar sample to each query has the same class label.
> Our proposed robust dataset distillation method yields an improvement of 3.6% to 4.2% on the top-1 accuracy.
> It indicates that the model trained with data distilled by RDD has better generalization capability.
>
> We have further refined **Section D.1** to include all experiments. We believe the inclusion of these more challenging and realistic evaluation settings can better illustrate the efficacy of the proposed method.
>
> ### **2. Choice of initialization [W2]**
>
> Thanks for the question. We generally adopt the same initialization of the baseline methods.
> For GLaD and IDC, the synthesized samples are randomly sampled from real images.
> We further conduct the experiment where the samples are initialized with clustering centers, which have a more evenly coverage over the original distribution.
> The results are shown in the table below. GLaD+RDD+Cluster indicates the initialization with clustering centers, which yields a similar performance as GLaD+RDD.
>
> | Dataset | IPC | GLaD | GLaD+RDD | GLaD+RDD+Cluster |
> | --- | --- | --- | --- | --- |
> | CIFAR-10 | 1 | 28.0 $\pm$ 0.8 | 29.2 $\pm$ 0.8 | 29.4 $\pm$ 0.9 |
> |  |              10 | 46.7 $\pm$ 0.6 | 50.2 $\pm$ 0.5 | 50.3 $\pm$ 0.6 |
> | ImageNet-10 | 1 | 33.5 $\pm$ 0.9 | 36.4 $\pm$ 0.8 | 36.5 $\pm$ 0.8 |
> | |                     10 | 50.9 $\pm$ 1.0 | 55.2 $\pm$ 1.1 | 55.0 $\pm$ 1.2 |
>
> Although random initialization cannot guarantee an even distribution at the beginning, during the subsequent optimization process, the algorithm is still robust enough to handle different initializations and provide a stable performance improvement over the baseline.
> We have added this part to **Section D.6** and **Table 11** in the revised appendix.

---

> > ### Author Response · Authors · 2024-11-21
> > **Response to Reviewer 1jHv (2/2)**
> >
> > ### **3. Experiments with other objectives [W3]**
> > This is an insightful comment. The method is applicable to all matching-based dataset distillation methods.
> > In Table 4 of the manuscript, we have included the experiment results on IDM, where distribution matching is adopted as the metric.
> > The results listed below suggest that our method also helps enhance performance for methods other than gradient matching.
> >
> > ### **4. Time complexity analysis [Q1]**
> > Thanks for the question.
> > Using CVaR instead of a sample average only takes a constant multiple of operations on the subsample. Clustering itself can be done polynomial in the number of samples. Since the properties of the model and loss function, that is, nonconvex and nonsmooth, are fundamentally unchanged, there is no change to the iteration complexity. We have added this part to **line 232** of the revised manuscript.
> >
> > In addition, we have included the computational time comparison between baseline and RDD in **Section D.3** of the appendix.  The baseline requires 70 seconds to finish a loop, while the CVaR loss calculation takes up an extra 30 seconds.
> > The robust optimization takes up less than 50% of the original calculation time.
> > The extra time is within an acceptable range, while providing significant improvement on the robustness.
> > We will make a clearer presentation in the revision. We will also aim to optimize the clustering implementation to further reduce the extra computational cost.
> >
> > ### **5. Notation clarification [Q2]**
> > Thank you for pointing out the issue.
> > This is the number of empirical samples taken, that corresponds practically to the synthetic and training data points.
> > We have accordingly added "samples of size $N$" to **line 271** of the revised manuscript.

---

> > > ### Comment · Reviewer_1jHv · 2024-11-25
> > > **Thank you for the detailed rebuttal**
> > >
> > > I appreciate the authors' detailed rebuttal. I will raise my rating.

---

> > > > ### Author Response · Authors · 2024-11-25
> > > >
> > > > Sincere thanks for recognizing the paper!
> > > >
> > > > We will further perform careful revision to address the comments and enhance the paper quality.
> > > >
> > > > Best regards,
> > > >
> > > > Authors

---

### Official Review · Reviewer_p8Rd · 2024-11-04

**Soundness:** 3
**Presentation:** 3
**Contribution:** 3
**Rating:** 6
**Confidence:** 3

**Summary:**

The work proposes a robust dataset distillation approach that incorporates distributional robust optimization (DRO) to enhance generalization and performance across subgroups. This method combines clustering with risk-minimized loss to conduct dataset distillation. By prioritizing representativeness and coverage over training error guarantees, the approach enhances the models trained on synthetic datasets for real-world scenarios. Both theoretical analysis and empirical validation on multiple standard benchmarks are provided, demonstrating the effectiveness of the proposed approach.

**Strengths:**

1. The paper proposes applying distributional robust optimization to dataset distillation, providing a reasonable approach to enhance generalization.

2. Drawing on distributional robust optimization theory, this work establishes a theoretical foundation to support the proposed approach to dataset distillation.

3. The paper is well-structured, with clear algorithm block and effective visualizations that enhance the presentation of the work.

**Weaknesses:**

1. The empirical experiments focus primarily on CIFAR-10, ImageNet-10. Extending the evaluation to larger, real-world datasets with a greater number of classes would better demonstrate the effectiveness of proposed approach in generalization and robustness under real-world conditions.

2. Comparing the proposed approach with additional baseline method addressing generalization and robustness would provide a more comprehensive assessment of the proposed approach, including comparisons with baseline methods such as [1, 2, 3].

3. It would be valuable to visualize the robust inference tasks using real-world data rather than illustrative visuals, which could provide more insight and observations in real-world scenarios.

Reference:

[1] Domain generalization with adversarial feature learning. CVPR 2018.

[2] Self-challenging Improves Cross-Domain Generalization. ECCV 2020.

[3] HYPO: Hyperspherical Out-of-Distribution Generalization. ICLR 2024.

**Questions:**

Please refer to the detailed suggestions provided in the weaknesses section.

---

> ### Author Response · Authors · 2024-11-21
> **Response to Reviewer p8Rd**
>
> Thank you for your detailed and constructive comments. Please find the responses below:
>
> ### **1. Experiment on datasets with more classes**
> Thanks for the question. In addition to CIFAR-10 and ImageNet-10, we also conduct the experiments on Tiny-ImageNet in Table 4. The proposed robust dataset distillation illustrates a 2% improvement over the baseline across all the listed metrics.
> The results indicate that the proposed method has the capability to be applied to even larger datasets and real-world applications to enhance the robustness of the distilled dataset.
>
> ### **2. Experiment on more domain generalization training methods**
> Thanks for the constructive advice.
> Accordingly, we apply these methods to the evaluation of the distilled dataset, and the results are listed in the table below.
>
> | Method | CIFAR-10 |  | ImageNet-10 | |
> | --- | --- | --- | --- | --- |
> | Vanilla | 46.7 $\pm$ 0.6 | **50.2** $\pm$ 0.5 | 50.9 $\pm$ 0.6 | **55.2** $\pm$ 1.1 |
> | MMD  | 47.1 $\pm$ 0.6 | **51.3** $\pm$ 0.5 | 51.8 $\pm$ 1.3 | **56.5** $\pm$ 1.2 |
> | RSC    | 47.9 $\pm$ 0.5 | **52.5** $\pm$ 0.5 | 52.4 $\pm$ 1.1 | **56.8** $\pm$ 1.0 |
> | HYPO | 49.0 $\pm$ 0.5 | **53.2** $\pm$ 0.6 | 53.6 $\pm$ 1.2 | **58.1** $\pm$ 1.1 |
>
> The results suggest that on the one hand, RDD consistently provides performance improvement on different training pipelines.
> On the other hand, the combination of RDD and other domain generalization methods can further improve the results over the baseline.
> We have added this part to **Section D.7** and **Table 12** in the revised appendix.
>
> ### **3. Experiment on more realistic settings**
> This is an insightful question. We agree that adding more realistic evaluation protocols can better illustrate the efficacy and practicality of our proposed method. And the proposed robust dataset distillation provides stable generalization improvement over the baseline.
>
> Initially, we have conducted a domain generalization experiment in **Section D.1** of the appendix. The model is first trained on one subset of ImageNet and evaluated on another subset through one-shot retrieval.
>
> Based on the comments of other reviewers, we further conduct experiments on domain transfer and subpopulation shift benchmarks.
>
> - In the first domain transfer experiment, the model is first trained on data distilled from a 50-class subset of ImageNet, and then fine-tuned on the fine-grained CUB-200 dataset.
>
> | Transfer Learning | GLaD | GLaD+RDD |
> | --- | --- | --- |
> | ImageNet-50 $\rightarrow$ CUB-200 | 49.2 $\pm$ 0.9 | **54.6** $\pm$ 0.6 |
>
> - In the second subpopulation shift experiment, the model is trained on the data distilled from the MetaShift benchmark, which involves spurious correlations. The results are shown below:
>
> | Metric | GLaD | GLaD+RDD |
> | --- | --- | --- |
> | Average Accuracy | 58.6 $\pm$ 2.3 | **62.2** $\pm$ 1.2 |
> | Worst-group Accuracy | 51.3 $\pm$ 1.8 ($\downarrow$ 7.3) | **57.0** $\pm$ 1.0 ($\downarrow$ 5.2) |
>
> The results of the above two experiments suggest that the proposed robust dataset distillation can be applied to a variety of real-world problems to enhance the robustness of the distilled data.
> We hope these results can address your concern. Please let us know if you have more specific settings that can illustrate the effectiveness of RDD.

---

> > ### Comment · Reviewer_p8Rd · 2024-11-25
> >
> > Thank you for the efforts. Most of my concerns have been addressed, and I would like to raise my score to 6.

---

> > > ### Author Response · Authors · 2024-11-25
> > >
> > > Thank you so much for your recognition!
> > >
> > > We believe that your comments have greatly helped us refine the paper. We will carefully revise the manuscript again to move some important results to the main text.
> > >
> > > Best regards,
> > >
> > > Authors

---

### Official Review · Reviewer_DaAx · 2024-11-04

**Soundness:** 2
**Presentation:** 2
**Contribution:** 2
**Rating:** 6
**Confidence:** 2

**Summary:**

This paper proposes an algorithm for dataset distillation by incorporating distributionally robust optimization into it. There is theoretical justification and empirical validation of the proposed algorithm.

**Strengths:**

There are both theoretical and experimental demonstrations of the effectiveness of the algorithm.

**Weaknesses:**

1. There seems to be a mismatch between the motivation and the experiments. The motivation emphasizes regions with low population density, which usually correspond to the worst-group performance in subpopulation shift [1]. However, the main experiments related to distribution shift are conducted on test sets with perturbations or the worst group induced by an additional clustering process. It would be better to conduct more experiments on subpopulation datasets included in [1].
2. The introduction has too many paragraphs, which makes the logic of the introduction tedious with poor readability.

Some minor issues:

- In Line 049, "some technique".
- In Line 129 and 131, "Algorithm" "Numerical Results" their first letters do not need to be capitalized.
- In the last paragraph of introduction, Section 3 is not mentioned.

[1] Yang, Yuzhe, et al. "Change is Hard: A Closer Look at Subpopulation Shift." *International Conference on Machine Learning*. PMLR, 2023.

**Questions:**

In Line 017, what does "targeting the training dataset" mean?

---

> ### Author Response · Authors · 2024-11-21
> **Response to Reviewer DaAx**
>
> Thank you for your detailed and constructive comments. Please find the responses below:
>
> ### **1.  Subpopulation shift experiment [W1]**
> Thanks for the valuable advice. We have accordingly conducted the dataset distillation experiments on MetaShift, which is included in [1]. After distillation, we used the simplest evaluation protocol consistent with that in the paper, such that the results are not influenced by other factors. The results are listed in the table below:
>
> | Metric | GLaD | GLaD+RDD |
> | --- | --- | --- |
> | Average Accuracy | 58.6 $\pm$ 2.3 | **62.2** $\pm$ 1.2 |
> | Worst-group Accuracy | 51.3 $\pm$ 1.8 ($\downarrow$ 7.3) | **57.0** $\pm$ 1.0 ($\downarrow$ 5.2) |
>
> As shown, the distilled data demonstrates substantially better performance, especially in the worst-group accuracy.
> We have included this part in the appendix in the revised manuscript in **Section D.1** and **Table 7**, which also contains some other challenging evaluation scenarios.
> We believe having these extra evaluation protocols better illustrates the efficacy of the proposed method in improving the robustness of dataset distillation algorithms.
>
> [1] Yang, Yuzhe, et al. "Change is Hard: A Closer Look at Subpopulation Shift." International Conference on Machine Learning. PMLR, 2023.
>
> ### **2.  Introduction paragraphs [W2]**
> Thanks for pointing out the defects. We have rewritten and substantially shortened the Introduction.
> If you still think the introduction is not sufficiently clear, please let us know.
>
> ### **3.  Minor issues [W3]**
> Thanks for reminding these mistakes. We have fixed the typos in the revision.
>
> ### **4.  Clarification of the abstract [Q1]**
> Thanks for pointing out the expression issue.
> We have corrected the original version to "using the empirical loss as the criterion" in the revised manuscript.

---

> > ### Comment · Reviewer_DaAx · 2024-11-25
> >
> > Thanks for your efforts! Considering the mismatch between the motivation (regions with low density) and experiments, I believe that the experiments of subpopulation shift should be conducted on more datasets with more algorithms, and they should be treated as the main experiments. Thus I decide to maintain my score, but I will not insist on rejection if all other reviewers champion acceptance.

---

> > > ### Author Response · Authors · 2024-11-25
> > >
> > > Thank you for your further comments.
> > >
> > > We agree that indeed the examples in [1] appear to be worthwhile to consider in more extensive detail. Accordingly, we conduct dataset distillation on another benchmark **ImageNetBG**, which focuses on the attribute generalization problem. We report the results in the table below:
> > >
> > > | Metric | GLaD | GLaD+RDD |
> > > | --- | --- | --- |
> > > | Average Accuracy | 41.7 $\pm$ 1.5 | **45.5** $\pm$ 1.1 |
> > > | Worst-group Accuracy | 32.2 $\pm$ 1.5 ($\downarrow$ 9.5) | **38.6** $\pm$ 1.0 ($\downarrow$ 6.9) |
> > >
> > > The results again suggest the effectiveness of the proposed robust dataset distillation method. Not only the average accuracy is improved, but the performance gap from the worst-group accuracy is also narrowed.
> > > We have revised the manuscript to add a new paragraph in the numerical results section (**line 407** and **Table 3**) to include the discussion on subpopulation shift benchmarks. We hope the new results can address your concern.
> > >
> > > Best regards,
> > >
> > > Authors

---

> > > > ### Comment · Reviewer_DaAx · 2024-11-29
> > > >
> > > > I appreciate authors' efforts in adding experiments. However, the reason for requiring these additional experiments of subpopulation shift is the mismatch between the motivation (regions with low density) and experiments. Although the added experiments are moved to the main paper, they are still not sufficient enough to match the motivation, considering the number of algorithms and datasets. Thus I still decide to maintain my score.

---

> > > > > ### Author Response · Authors · 2024-12-03
> > > > >
> > > > > Dear reviewer DaAx,
> > > > >
> > > > > We further conduct the experiment on another benchmark **CelebA**. The results are listed in the table below.
> > > > >
> > > > > | Dataset | CelebA | | | |
> > > > > | --- | --- | --- | --- | --- |
> > > > > | | GLaD | GLaD+RDD | IDC | IDC+RDD |
> > > > > | Average Accuracy | 33.5 $\pm$ 3.0 | **36.9** $\pm$ 2.8 | 40.3 $\pm$ 2.8 | **41.6** $\pm$ 2.5 |
> > > > > | Worst-group Accuracy | 21.2 $\pm$ 3.5 ($\downarrow$ 12.3) | **27.8** $\pm$ 2.6 ($\downarrow$ 9.1) | 29.7 $\pm$ 3.0 ($\downarrow$ 10.6) | **32.3** $\pm$ 2.5 ($\downarrow$ 9.3) |
> > > > >
> > > > > The results again suggest the effectiveness of the proposed robust dataset distillation method. We are also running the experiments on **Living17**. We will update the results if we can meet the discussion deadline. Otherwise, we promise we will update the results in the manuscript. We hope these new results can address your concern about the mismatching between the motivation and experimental results. If you have further advice on presenting these results, given that it is the last day that reviewers can participate in the discussion, please do let us know.
> > > > >
> > > > > Sincerely,
> > > > >
> > > > > Authors

---

> > > > > > ### Comment · Reviewer_DaAx · 2024-12-03
> > > > > >
> > > > > > Thanks for your response and efforts in adding experiments. I would increase my rating accordingly.

---

> > > > > > > ### Author Response · Authors · 2024-12-03
> > > > > > >
> > > > > > > Thank you so much for providing invaluable comments to improve the paper quality. We will carefully revise the manuscript to add these new experimental results.
> > > > > > >
> > > > > > > Thanks again for recognizing the effectiveness of the RDD method!
> > > > > > >
> > > > > > > Best regards,
> > > > > > >
> > > > > > > Authors

---

> ### Author Response · Authors · 2024-12-01
>
> Dear reviewer DaAx,
>
> Thank you for your further comment. In addition to GLaD, we also conduct the experiment on the other main baseline in the paper, IDC. The results are listed in the table below.
>
> | Dataset | MetaShift | | ImageNetBG | |
> | --- | --- | --- | --- | --- |
> | | IDC | IDC+RDD | IDC | IDC+RDD |
> | Average Accuracy | 69.7 $\pm$ 1.9 | **72.1** $\pm$ 1.8 | 60.2 $\pm$ 1.0 | **62.7** $\pm$ 0.9 |
> | Worst-group Accuracy | 62.8 $\pm$ 1.9 ($\downarrow$ 6.9) | **67.0** $\pm$ 1.9 ($\downarrow$ 5.1) | 51.6 $\pm$ 1.2 ($\downarrow$ 8.6) | **56.0** $\pm$ 1.0 ($\downarrow$ 6.7) |
>
> The data distilled by IDC illustrates a much better overall performance than that of GLaD. Yet, the proposed robust dataset distillation method still achieves considerable improvement over IDC on these two datasets.
>
> We also want to explain why we conduct experiments on these two benchmarks. Following the previous dataset distillation setting, for each class in a dataset, we distill the same amount of samples, which will break the class imbalance in some subpopulation shift benchmarks. Therefore, we select these two benchmarks that do not have the class imbalance shift. Nevertheless, we conduct a standard distillation experiment on the Waterbirds benchmark, which contains three types of shifts including class imbalance, and the results are listed in the table below.
>
> | Dataset | Waterbirds | | | |
> | --- | --- | --- | --- | --- |
> | | GLaD | GLaD+RDD | IDC | IDC+RDD |
> | Average Accuracy | 51.6 $\pm$ 2.3 | **55.6** $\pm$ 1.2 | 59.6 $\pm$ 2.0 | **61.3** $\pm$ 1.9 |
> | Worst-group Accuracy | 40.3 $\pm$ 1.8 ($\downarrow$ 10.3) | **47.9** $\pm$ 1.0 ($\downarrow$ 7.7) | 50.0 $\pm$ 1.7 ($\downarrow$ 9.6) | **53.5** $\pm$ 1.5 ($\downarrow$ 7.8) |
>
> The results show that the baseline GLaD cannot effectively handle the problem, whose performance is only slightly higher than random guessing. By applying RDD, the method obtains considerable performance improvement on both average accuracy and worst-group accuracy.
>
> We will include these new results in the paper. We hope these new results can address your concern about the effectiveness of the proposed RDD method on subpopulation shift benchmarks. If you have further suggestions on how to better illustrate the effectiveness on subpopulation shift benchmarks, please let us know. Even if we cannot finish the experiments before the rebuttal deadline, we promise we will include all the results in the paper.
>
> Best regards,
>
> Authors

---

### Author Response · Authors · 2024-11-21
**General Response**

We want to express sincere gratitude to the reviewers for their detailed and constructive comments and advice on the manuscript.
We are grateful that all reviewers acknowledge our theoretical analysis of the connection between distributionally robust optimization and dataset distillation robustness, as well as the experimental results to support the effectiveness of the proposed method.
We have carefully revised the manuscript and marked the modified parts as **blue**, with the mark "**NEW**" on the right of the modified contents.

Specifically, we have made the following major revisions:

1. We have included more realistic evaluation scenarios in **Section D.1** of the revised appendix. Now the section contains:

- Domain transfer from pre-training to downstream fine-tuning on fine-grained classification.

- Domain generalization from one ImageNet subset to another subset.

- Subpopulation shift benchmark involving Spurious Correlations.

2. We have included the comparison on more domain generalization training methods in **Section D.7**.

3. We have added the ablation study on the initialization of synthetic samples in **Section D.6**, where the proposed RDD method is robust enough to handle different initialization.

4. We have rewritten some parts of the manuscript to enhance the clarity of our idea and fixed some vague expressions.

We hope that these modifications have addressed the concerns raised by the reviewers.
We want to thank the reviewers again for their insightful opinions that help further refine this paper. We welcome any feedback and discussion from you. If you fell our response cannot address your concerns, please kindly let us know.

---

### Meta-Review · Area_Chair_wSay · 2024-12-18

**Metareview:**

The submission develops a novel formulation of dataset distillation that makes use of ideas from distributionally robust optimisation. A method based on segmenting the input space into different subgroups and ensuring that a distilled dataset leads to a model that has good worst-case performance on these subgroups is provided. The reviewers agree that this direction is interesting, novel, and well-executed. However, all reviewers had some concerns about the particular datasets distribution shifts used in the experimental validation of the new method.

I agree with the consensus the reviewers have come to; despite some issues about realism of the experimental evaluation, the paper makes a good contribution and should be accepted.

**Additional Comments On Reviewer Discussion:**

The main issue discussed was related to the data used for experiments, which was resolved by the introduction of new experimental results. This further increased my confidence in accepting the paper.

---

### Decision · Program_Chairs · 2025-01-22

Accept (Poster)